# A Two-Stage Multi-Objective Genetic Algorithm for a Flexible Job Shop Scheduling Problem with Lot Streaming

**Danial Rooyani** 🆔 and **Fantahun Defersha** *🆔

School of Engineering, University of Guelph, Guelph, ON N1G 2W1, Canada; drooyani@uoguelph.ca
* Correspondence: fdefersh@uoguelph.ca; Tel.: +1-519-824-4120 (ext. 56512)

**Abstract:** The work in this paper is motivated by a recently published article in which the authors developed an efficient two-stage genetic algorithm for a comprehensive model of a flexible job-shop scheduling problem (FJSP). In this paper, we extend the application of the algorithm to solve a lot streaming problem in FJSP while at the same time expanding the model to incorporate multiple objectives. The objective function terms included in our current work are the minimization of the (1) makespan, (2) maximum sublot flowtime, (3) total sublot flow time, (4) maximum job flowtime, (5) total job flow time, (6) maximum sublot finish-time separation, (7) total sublot finish-time separation, (8) maximum machine load, (9) total machine load, and (10) maximum machine load difference. Numerical examples are presented to illustrate the greater need for multi-objective optimization in larger problems, the interaction of the various objective function terms, and their relevance in providing better solution quality. The ability of the two-stage genetic algorithm to jointly optimize all the objective function terms is also investigated. The results show that the algorithm can generate initial solutions that are highly improved in all of the objective function terms. It also outperforms the regular genetic algorithm in convergence speed and final solution quality in solving the multi-objective FJSP lot streaming. We also demonstrate that high-performance parallel computation can further improve the performance of the two-stage genetic algorithm. Nevertheless, the sequential two-stage genetic algorithm with a single CPU outperforms the parallel regular genetic algorithm that uses many CPUs, asserting the superiority of the two-stage genetic algorithm in solving the proposed multi-objective FJSP lot streaming.

**Keywords:** flexible job shop scheduling; lot streaming; multi-objective optimization; two-stage genetic algorithm; parallel computation

## 1. Introduction

Defersha and Rooyani [1] developed an efficient two-stage genetic algorithm for a comprehensive flexible job shop scheduling problem (FJSP) that incorporates (1) sequence-dependent setup time, (2) attached and detached nature of setup, (3) machine release date, and (4) lag-time. The high performance of the two-stage algorithm was achieved by a systematically designed solution representation and a greedy decoding mechanism of the first stage. The approach enabled the algorithm to find highly improved solutions from the start and rapidly converge to promising regions of the search space. The second stage removes the greedy nature of the first stage by following the regular approach of a genetic algorithm for FJSP and attempts to improve the solutions found in the first stage. The authors demonstrated the superiority of the developed two-stage genetic algorithm in solving large-size problems with up to 80 machines and 140 jobs.

In this paper, we extend the application of the algorithm in [1] to solve a lot streaming problem in FJSP that appeared in [2] while, at the same time, expanding the problem to incorporate multiple objective functions. The objective function terms included are the minimization of the (1) makespan, (2) maximum sublot flowtime, (3) total sublot flow time, (4) maximum job flowtime, (5) total job flow time, (6) maximum sublot finish-time

separation, (7) total sublot finish-time separation, (8) maximum machine load, (9) total machine load, and (10) maximum machine load difference.

In addition to expanding the single objective FJSP lot streaming to a multi-objective one and customizing the two-stage genetic algorithm to solve it, an added contribution of this paper is its provision of many numerical studies. At the outset of the numerical studies, all the ten objective function terms are illustrated using a small prototype problem. The importance of multi-objective optimization in small versus large size problems is examined and contrasted. The capability of the two-stage genetic algorithm to jointly optimize all the objective function terms is evaluated. The need to optimize both the maximum and the total of a performance measure (such as flowtime) is examined.

The relevance of two newly proposed objective function terms (sublot finish-time separation and maximum workload difference) in providing better solution quality is assessed. The quality of the initial population and the convergence behavior of the two-stage genetic algorithm is contrasted against the regular genetic algorithm with respect to each of the objective function terms. Further algorithm enhancement through high-performance parallel computation is considered. Algorithm components and parameters are empirically studied. In particular, three different selection operators are examined, and an Analysis of Variance (ANOVA) on mutation and crossover probabilities is conducted.

The remainder of this article is organized as follows. In Section 2, recent articles in lot streaming are reviewed. The proposed multi-objective FJSP lot streaming model is presented in Section 3. The adaptation of the two-stage genetic algorithm to solve the multi-objective lot streaming model is detailed in Section 4. Section 5 provides extensive numerical studies. Our conclusions, discussion, and future research are in Section 6.

## 2. Literature Review

Lot streaming is a technique that splits a production lot of a job into several independent sublots and allows a sublot to be transferred from one machine to the next without waiting for the other sublots. In doing so, it enables the simultaneous processing of the sublots of a given job on multiple machines, thereby, reducing the completion time of the job. The approach has been used as a strategy for a time-based competition in today's global market [3]. Since its formal introduction in [4], it has been an active topic of research, and many articles have been published in its application to scheduling in a variety of shop configurations. The following subsections briefly review recent articles on lot streaming based on those shop configurations. Comprehensive reviews of publications on lot streaming can be found in [3,5].

### 2.1. Pure Flow Shop Lot Streaming (PFS-LS)

The majority of early publications on lot streaming are for pure flow shops. However, recent literature indicates that PFS-LS still continues to attract the attention of the research community. A tabu-search based three-stage algorithm for PFS-LS was developed in [6]. The three stages of the algorithm involve (i) predetermining sublot sizes, (ii) developing a schedule based on the predetermined sublot sizes, and (iii) varying the sizes of the sublots to improve the solution quality. Defersha and Chen [7] developed a linear programming hybridized genetic algorithm with variable sublots to minimize the makespan. The authors demonstrated that, in the presence of setup time, variable sublot could bring substantial improvement in makespan compared to consistent or equal-sized sublots.

A genetic algorithm for PFS-LS with limited buffer capacities and equal-sized sublots was developed in [8] to minimize earliness and tardiness. Han et al. [9] developed a multi-objective genetic algorithm (given the number and size sublots) to minimize the makespan, total flow time, machine idle time, and earliness time. Meng et al. [10] developed an improved migrant birds optimization for minimizing the makespan with equal sublots and sequence-dependent setup time. A bee colony algorithm was proposed in [11] to minimize the makespan and earliness in a blocking PFS-LS with no intermediate buffer between

adjacent stages. The author combined setup time with processing time and assumed that the number and size of sublots are determined before scheduling.

An exact heuristic based on dynamic programming and Lagrangian relaxation was developed in [12] for a two-machine PFS-LS to minimize total flowtime. A convex programming technique for a single-job two-machine PFS-LS was developed in [13] with due date criterion and minimization of the total energy consumption. The energy consumption was optimized by varying the processing speed of the sublots. Wang et al. [14] developed an algorithm for PFS-LS with intermingling and variable sublots having detached setups. The authors demonstrated that the assumption of a detached setup could reduce the makespan.

### 2.2. Hybrid Flow Shop Lot Streaming (HFS-LS)

Early and some recent publications in HFS-LS (e.g., [15–20]) are limited to a special case where there are only two stages. To the best of our knowledge, the first major research effort in lot streaming in the general hybrid flexible flow shop with more than two stages was reported in [21]. The authors developed a parallel genetic algorithm with makespan criterion, sequence-dependent setup time, and machine release date. The authors also demonstrated that lot streaming could bring greater makespan reduction in hybrid flows shop than in pure flow shop as the former allows the overlapping of operations not only across stages but also within the parallel machines of a given stage. Nejati et al. [22] developed a genetic algorithm for HFS-LS in the presence of work-shift constraint. The authors assumed that the processing of a sublot cannot be started if the remaining time of the work-shift does not allow the sublot to be completed, in which case, the sublot has to wait for the next work-shift.

Techniques based on migrant birds optimization were developed in [23,24] to minimize the total flow time and makespan, respectively. Chen et al. [25] proposed a genetic algorithm to minimize the makespan and energy utilization. The minimization of energy utilization is achieved via machine selection, where each stage may have unrelated parallel machines with different power consumption and processing speed. Energy-aware multi-objective HFS-LS was presented in [26] to minimize the average sojourn time, energy consumption, earliness, and tardiness. Zhang et al. [27] developed an evolutionary algorithm (with consistent sublots) to minimize the makespan and number of sublots in the presence of setup and transportation.

### 2.3. Classical Job Shop Lot Streaming (CJS-LS)

The first paper in lot streaming (i.e., [4]) was for a classical job shop scheduling problem. However, research in CJS-LS is minimal. Hereunder, we reviewed relatively recent articles in the area. Chan et al. [28] and Wong et al. [29] considered CJS-LS with due date criterion, where the authors assumed that all the sublots and the jobs from the same product (with a due date) will be assembled at the end of the line. Methodologies based on a generic algorithm were developed to minimize the total cost of earliness, lateness, and setup in [30,31].

The authors stated that excessive lot splitting could increase the setup cost. A CJS-LS problem, where a customer order contains several jobs and shipment can happen only when all the sublots of the jobs of a given order are completed, was considered in [32]. The authors developed a genetic algorithm to solve the considered problem to minimize the makespan, lateness, and flowtime of the finished goods. Liu et al. [33] developed methodologies to maximize the total values of the jobs. The authors assumed that the value of a job deteriorates exponentially over time, and the sooner the job completes, the higher its value is. Lei and Guo [34] developed a bee colony algorithm with makespan criterion in the presence of a single transporter that could transfer one sublot at a time.

### 2.4. Flexible Job Shop Lot Streaming (FJS-LS)

Early research in job shop lot streaming was for the classical job shop. However, in recent years, the research focus on job shop lot streaming has been in FJS-LS. Defersha

and Chen [2] developed a parallel genetic algorithm for FJS-LS with makespan criterion considering sequence-dependent setup time, attached and detached nature of setups, machine release date, and lag-time (a delay for cooling, drying, inspection, or other ancillary operations). Demir and Işleyen [35] and Meng et al. [36] implemented FJS-LS through a successive partial transfer of a job from one machine to the next to allow operation overlapping when sublots are not scheduled independently. Bożek and Werner [37] considered FJS-LS with variable sublots in a two-stage approach.

In the first stage, the makespan is minimized with the minimum sizes of the sublots defined for the problem (the larger number of sublots). In the second stage, the number of the sublots is reduced without affecting the makespan to minimize the transportation cost. Defersha and Bayat Movahed [38] developed a linear programming hybridized genetic algorithm where the linear programming is periodically used to enhance promising solutions during the search process. Novas [39] developed a method based on constraint programming to solve flexible job shop lot streaming with makespan criterion.

Daneshamooz et al. [40] proposed an algorithm based on a variable neighborhood search to minimize the makespan for a lot streaming problem in a flexible job shop followed by a parallel assembly station. An FJS-LS problem in an Engineer to order environment was presented in [41]. The authors developed a mathematical model with variable sublot and makespan criterion and then proposed a genetic-algorithm-based heuristic to solve the model effectively. Though the above review indicates momentum in FJS-LS research, the total number of publications is minimal, and further research needs to be conducted.

## 3. Mathematical Modeling

### 3.1. The Basic Problem

The main objective of this paper is to expand the single objective FJSP lot streaming model presented in [2] to a multi-objective one and develop a two-stage genetic-algorithm-based on the work in [1]. However, for better comprehension of this paper, we first present the basic single-objective problem and its mathematical model as presented in [2].

### 3.1.1. Problem Description and Notations

Consider a job shop consisting of $M$ machines where machines with common functionalities are grouped into a department (e.g., turning machines in a turning department). Assume that the system is currently processing jobs from the previous schedule, and each machine $m$ (where $m = 1, \ldots, M$) has a release date $D_m$—at which time, it will be available for the next scheduling. Consider also a total of $J$ independent jobs to be scheduled next in the system where a job is a batch of identical parts. The number of parts in a batch of job $j$ (where $j = 1, \ldots, J$) is given by $B_j$, and this batch is to be split into $S_j$ number of unequal sublots (transfer batches).

A decision variable $b_{s,j}$ is used to denote the size of sublot $s$ (where $s = 1, \ldots, S_j$) of job $j$. Each sublot of job $j$ is to undergo $O_j$ number of operations in a fixed sequence such that each operation $o$ (where $o = 1, \ldots, O_j$) can be processed by one of several eligible machines. $T_{o,j,m}$ is unit-processing-time for operation $o$ of job $j$ on machine $m$. The operation $o$ of a sublot of job $j$ can be started on an eligible machine $m$ after lag time $L_{o,j}$ and after the setup is performed. The lag time $L_{o,j}$ is a waiting time that may be required either for cooling, drying, or for some other purpose. The setup time for an operation $o$ of job type $j$ on machine $m$ depends on the preceding operations and is denoted by $S_{o,j,m,o',j'}$, where operation $o'$ of a sublot of job $j'$ is the preceding operation on machine $m$.

If operation $o$ of sublot $s$ of job $j$ is the first operation to be processed on machine $m$, the setup time is represented as $S_{o,j,m}^*$. The setup time $S_{o,j,m,o',j'}$ (or $S_{o,j,m}^*$) for operation $o$ of a sublot of job $j$ can be overlapped with the processing time of operation $o - 1$ of the same sublot if the setup is a detached setup and machine $m$ is available for setup. The problem is to determine the size of each sublot, assign the operation of each sublot to one of the eligible machines and determine the sequence and starting time of the assigned operations on each machine. The objective is to minimize the makespan of the schedule. We next

introduce some additional notations and then present a mixed-integer linear programming (MILP) formulation for FJSP-LS.

Additional Parameters:

$R_m$    The maximum number of production runs of machine $m$ where production runs are indexed by $r = 1, 2, \ldots, R_m$. Each of these production runs can be assigned to, at most, one operation of one sublot. Thus, the assignment of the operations to production runs of a given machine determines the sequence of the operations on that machine.

$P_{o,j,m}$    A binary data point equal to 1 if operation $o$ of job $j$ can be processed on machine $m$, and 0 otherwise.

$A_{o,j}$    A binary data point equal to 1 if the setup of operation $o$ of of job $j$ is attached (non-anticipatory), or 0 if this setup is detached (anticipatory).

$\Omega$    Large positive number.

Variables:

*Continuous Variables:*

$c_{max}$    Makespan of the schedule.

$c_{o,s,j}$    Completion time of operation $o$ of sublot $s$ of job $j$.

$\widehat{c}_{r,m}$    Completion time of the $r$th run of machine $m$.

$b_{s,j}$    Size of sublot $s$ of job $j$.

*Binary Integer Variables:*

$x_{r,m,o,s,j}$    A binary variable that takes the value 1 if the $r$th run on machine $m$ is for operation $o$ of sublot $s$ of job $j$, and 0 otherwise.

$y_{r,m,o,j}$    A binary variable that takes the value 1 if the $r$th run on machine $m$ is for operation $o$ of any one of the sublots of job $j$, and 0 otherwise.

$\gamma_{s,j}$    A binary variable that takes the value 1 if sublot $s$ of job $j$ is non-zero ($b_{s,j} \geq 1$), and 0 otherwise.

$z_{r,m}$    A binary variable that takes the value 1 if the $r$th potential run of machine $m$ has been assigned to an operation, and 0 otherwise.

### 3.1.2. MILP Model for FJSP-LS

Following the problem description and using the notations given above, the MILP mathematical model for the FJSP-LS is presented below.

Minimize:

$$Objective = c_{max} \tag{1}$$

Subject to:

$$c_{max} \geq c_{o,s,j} \; ; \;\; \forall(o, s, j) \tag{2}$$

$$\widehat{c}_{r,m} \geq c_{o,s,j} + \Omega \cdot x_{r,m,o,s,j} - \Omega \; ; \;\; \forall(r, m, o, s, j) \tag{3}$$

$$\widehat{c}_{r,m} \leq c_{o,s,j} - \Omega \cdot x_{r,m,o,s,j} + \Omega \; ; \;\; \forall(r, m, o, s, j) \tag{4}$$

$$\widehat{c}_{1,m} - b_{s,j} \cdot T_{o,j,m} - S^*_{o,j,m} - \Omega \cdot x_{1,m,o,s,j} + \Omega \geq D_m ; \quad \forall (m,o,s,j) \tag{5}$$

$$\widehat{c}_{r,m} - b_{s,j} \cdot T_{o,j,m} - S_{o,j,m,o',j'} - \Omega \cdot (y_{r-1,m,o',j'} + x_{r,m,o,s,j}) + 2\Omega \geq \widehat{c}_{r-1,m} ;$$
$$\forall (r,m,o,s,j,o',j') | (r > 1) \tag{6}$$

$$\widehat{c}_{1,m} - b_{s,j} \cdot T_{o,j,m} - S^*_{o,j,m} \cdot A_{o,j} - \Omega \cdot (x_{1,m,o,s,j} + x_{r',m',o-1,s,j}) + 2\Omega \geq \widehat{c}_{r',m'} + L_{o,j};$$
$$\forall (m,r',m',o,s,j) | \{ ((1,m) \neq (r',m')) \wedge (o > 1) \} \tag{7}$$

$$\widehat{c}_{r,m} - b_{s,j} \cdot T_{o,j,m} - S_{o,j,m,o',j'} \cdot A_{o,j} - \Omega \cdot (y_{r-1,m,o',j'} + x_{r,m,o,s,j} + x_{r',m',o-1,s,j}) + 3\Omega$$
$$\geq \widehat{c}_{r',m'} + L_{o,j} ; \quad \forall (r,m,r',m',o,s,j,o',j') |$$
$$\{ (r > 1) \wedge (o > 1) \wedge (r,m) \neq (r',m') \wedge (o,j) \neq (o',j') \} \tag{8}$$

$$y_{r,m,o,j} \leq P_{o,j,m} ; \quad \forall (r,m,o,j) \tag{9}$$

$$y_{r,m,o,j} = \sum_{s=1}^{S_j} x_{r,m,o,s,j} ; \quad \forall (r,m,o,j) \tag{10}$$

$$\sum_{m=1}^{M} \sum_{r=1}^{R_m} x_{r,m,o,s,j} = \gamma_{s,j} ; \quad \forall (o,s,j) \tag{11}$$

$$b_{s,j} \leq B_j \cdot \gamma_{s,j} ; \quad \forall (s,j) \tag{12}$$

$$\gamma_{s,j} \leq b_{s,j} ; \quad \forall (s,j) \tag{13}$$

$$\sum_{s=1}^{S_j} b_{s,j} = B_j ; \quad \forall (j) \tag{14}$$

$$\sum_{j=1}^{J} \sum_{s=1}^{S_j} \sum_{o=1}^{O_j} x_{r,m,o,s,j} = z_{r,m} ; \quad \forall (r,m) \tag{15}$$

$$z_{r+1,m} \leq z_{r,m} ; \quad \forall (r,m) \tag{16}$$

$$x_{r',m,o',s,j} \leq 1 - x_{r,m,o,s,j}; \quad \forall (r,r',m,o,o',s,j) | \{ (o' > o) \wedge (r' < r) \} \tag{17}$$

$$x_{r',m,o',s,j} \leq 1 - x_{r,m,o,s,j}; \quad \forall (r,r',m,o,o',s,j) | \{ (o' < o) \wedge (r' > r) \} \tag{18}$$

$$x_{r,m,o,s,j}, \; y_{r,m,o,j}, \; \gamma_{s,j} \text{ and } z_{r,m} \text{ are binary} \tag{19}$$

The complete description and the meanings of the objective function in Equation (1) and the constraints in Equations (2)–(19) can be found in [2].The expansion of this single objective FJSP lot streaming model into a multi-objective one is presented in the following section.

### 3.2. Multi-Objective Model for FJSP-LS

As it was stated previously, one of the objectives of this paper is to expand the single objective FJSP lot streaming presented in [2] to a multi objective approach. In this section, we present notations of additional continuous variables and the MILP formulation of the proposed multi-objective FJSP scheduling with lot streaming.

### 3.2.1. Additional Continuous Variables

The definitions of the additional continuous variables is given below. Further explanations for some of the variable definitions are also given as we discus the equations that use those variables.

| | |
|---|---|
| $e_{s,j}$ | Entry time of sublot $s$ of job $j$. |
| $\hat{e}_j$ | Entry time of job $j$ (minimum of $e_{s,j}$ for all $s$ of job $j$). |
| $d_{s,j}$ | Departure time of sublot $s$ of job $j$. |
| $\hat{d}_j$ | Departure time of job $j$ (maximum of $d_{s,j}$ for all $s$ of job $j$). |
| $f_{s,j}$ | Flowtime of sublot $s$ of job $j$. |
| $\hat{f}_j$ | Flowtime of job $j$. |
| $f_{max}$ | Maximum sublot flowtime. |
| $\hat{f}_{max}$ | Maximum job flowtime. |
| $f_{total}$ | Total sublot flowtime. |
| $\hat{f}_{total}$ | Total job flowtime. |
| $\hat{g}_j$ | Minimum sublot departure time of job $j$. |
| $\hat{h}_j$ | Sublot finish separation time of job $j$. |
| $\hat{h}_{max}$ | Maximum sublot finish separation time. |
| $\hat{h}_{total}$ | Total sublot finish separation time. |
| $l_{m,o,s,j}$ | Workload on machine $m$ because of the setup and processing of operation $o$ of sublot $s$ of job $j$. |
| $\hat{l}_m$ | Workload on machine $m$. |
| $\hat{l}_{min}$ | Minimum machine workload. |
| $\hat{l}_{max}$ | Maximum machine workload. |
| $\hat{l}_{total}$ | Total machine workload. |
| $\hat{l}_{diff}$ | Maximum machine workload difference. |

### 3.2.2. Objective Functions and Additional Constraints

The objective of the proposed multi-objective model is to minimize the function given in Equation (20) subject to the constraints in the original model in Equations (2)–(19) and newly added constraints in Equations (21)–(56). The objective function terms and the additional constraints are discussed in the following sections.

$$\text{Minimize: } Z_i \ \forall i \in \{1, 2, \cdots, 10\}. \tag{20}$$

### 3.2.3. Makespan ($Z_1$)

The makespan is defined as the maximum completion time of a given schedule. Its minimization is a widely used objective function in scheduling research. The essence of minimizing the makespan is to finish production as soon as possible to expedite the delivery of products to customers and/or to quickly free up resources for the upcoming production and other tasks, such as development and maintenance. The first objective function ($Z_1$) of the proposed multi-objective model is makespan ($c_{max}$) as shown in Equation (21).

$$Z_1 = c_{max} \tag{21}$$

### 3.2.4. Maximum and Total Sublot Flowtime ($Z_2$ and $Z_3$)

The entry time ($e_{s,j}$) to the shop floor of a sublot of a job is the time the setup of its first operation begins if the setup is attached. If the setup of the first operation is detached, $e_{s,j}$ is the time at which the actual processing of the first operation begins as setup can be completed before the raw material is admitted to the shop floor. The constraints in Equations (22)–(25) are used to set the value of this variable. The departure time of the sublot ($d_{s,j}$) is simply the completion time of the last operation of the sublot as enforced by the constraint in Equation (26).

The flowtime of sublot $s$ of job $j$, denoted as $f_{s,j}$, is the interval between the time the sublot enters the shop floor to the time its last operation is finished. Its value is set by the constraint in Equation (27). The constraint in Equation (28) along with the objective function will enforce $f_{max}$ to assume the maximum flowtime of all the sublots ($\max_{\forall(s,j)} f_{s,j}$). The total flowtime of all the sublots is calculated by the constraint in Equation (29). The objective function terms $Z_2$ and $Z_3$ are the values of $f_{max}$ and $f_{total}$ as shown in Equations (30) and (31), respectively. The minimization of flowtime can lead to stable or uniform utilization of resources and a rapid turn-around of jobs, and it is particularly important in real-life situations where reducing inventory or holding cost is of primary concern [42].

$$e_{s,j} \geq c_{1,s,j} - b_{s,j} \cdot T_{1,j,m} - S^*_{1,j,m} \cdot A_{1,j} - \Omega \cdot (1 - x_{1,m,1,s,j}); \quad \forall(s,j,m) \tag{22}$$

$$e_{s,j} \leq c_{1,s,j} - b_{s,j} \cdot T_{1,j,m} - S^*_{1,j,m} \cdot A_{1,j} + \Omega \cdot (1 - x_{1,m,1,s,j}); \quad \forall(s,j,m) \tag{23}$$

$$e_{s,j} \geq c_{1,s,j} - b_{s,j} \cdot T_{1,j,m} - S_{1,j,m,o',j'} \cdot A_{1,j} - 2\Omega \cdot (1 - x_{r,m,1,s,j} - y_{r-1,m,o',j'});$$
$$\forall(s,j,r,m)|r>1 \tag{24}$$

$$e_{s,j} \leq c_{1,s,j} - b_{s,j} \cdot T_{1,j,m} - S_{1,j,m,o',j'} \cdot A_{1,j} + 2\Omega \cdot (1 - x_{r,m,1,s,j} - y_{r-1,m,o',j'});$$
$$\forall(s,j,r,m)|r>1 \tag{25}$$

$$d_{s,j} = c_{O_j,s,j}; \quad \forall(s,j) \tag{26}$$

$$f_{s,j} = d_{s,j} - e_{s,j}; \quad \forall(s,j) \tag{27}$$

$$f_{max} \geq f_{s,j}; \quad \forall(s,j) \tag{28}$$

$$f_{total} = \sum_{j=1}^{J} \sum_{s=1}^{S_j} f_{s,j} \tag{29}$$

$$Z_2 = f_{max} \tag{30}$$

$$Z_3 = f_{total} \tag{31}$$

### 3.2.5. Maximum and Total Job Flowtime ($Z_4$ and $Z_5$)

In the presence of lot streaming, the entrance time $\hat{e}_j$ and the departure time $\hat{d}_j$ of a job are the smallest and the largest entrance times of all its sublots, $\min_{\forall s|\gamma_{s,j}=0}\{e_{s,j}\}$ and $\max_{\forall s|\gamma_{s,j}=0}\{d_{s,j}\}$, respectively. The values of these variables are set by the constraints in Equations (32) and (33), and the objective function. The flowtime of a job, $\hat{f}_j$, is the difference $\hat{d}_j - \hat{e}_j$ as enforced by the constraint in Equation (34). The constraint in Equation (35) along the objective function enforces $\hat{f}_{max}$ to assume the maximum flowtime of all the jobs, $\max_{\forall j}\{\hat{f}_j\}$. The total job flowtime ($\hat{f}_{total}$) is evaluated by the constraint in Equation (36). The values $\hat{f}_{max}$ and $\hat{f}_{total}$ correspond to the fourth and fifth terms, $Z_4$ and $Z_5$, of the objective function and their values are enforced by the constraints in Equations (37) and (38), respectively.

$$\hat{e}_j \leq e_{s,j} + \Omega(1 - \gamma_{s,j}); \quad \forall(s,j) \tag{32}$$

$$\hat{d}_j \geq d_{s,j} - \Omega(1 - \gamma_{s,j}); \quad \forall(s,j) \tag{33}$$

$$\hat{f}_j = \hat{d}_j - \hat{e}_j; \quad \forall j \tag{34}$$

$$\hat{f}_{max} \geq \hat{f}_j; \quad \forall j \tag{35}$$

$$\hat{f}_{total} = \sum_{j=1}^{J} \hat{f}_j \tag{36}$$

$$Z_4 = \hat{f}_{max} \tag{37}$$

$$Z_5 = \hat{f}_{total} \tag{38}$$

### 3.2.6. Maximum and Total Sublot Finish-Time Separation ($Z_6$ and $Z_7$)

In lot streaming, sublots are treated independently. As a result, one sublot of a job may be finished much sooner than the other sublot of the same job. This may increase work-in-process inventory as the entire job can not be made available for shipment or assembly within a reasonable time window. Hence, in this research, we introduce an objective function to minimize the gap between the earliest and the latest finish-times of sublots of the same job. In doing so, first we defined a variable $\hat{g}_j$ that assumes the earliest finish-time among all the sublots of a job, $\min_{\forall(s,j)|\gamma_{s,j}=1}\{d_{s,j}\}$, as enforced by the constraint in Equation (39) and the objective function.

The latest finish-time of the sublots of a job is its departure time $\hat{d}_j$, which was discussed previously. With these variables defined, the sublot finish separation time of a job, $\hat{h}_j$, is the difference $\hat{d}_j - \hat{g}_j$, enforced by the constraint in Equation (40). The constraint in Equation (41) and the objective function will enforce $\hat{h}_{max}$ to assume the value $\max_{\forall j}\{\hat{h}_j\}$. The total sublot finish time separation $\hat{h}_{total}$ is evaluated using the constraint in Equation (42). The objective function terms $Z_6$ and $Z_7$ correspond to the values of $\hat{h}_{max}$ and $\hat{h}_{total}$, respectively, as enforced by the constraints in Equations (43) and (44).

$$\hat{g}_j \leq d_{s,j} + \Omega(1 - \gamma_{s,j}); \quad \forall(s,j) \tag{39}$$

$$\hat{h}_j = \hat{d}_j - \hat{g}_j; \quad \forall(j) \tag{40}$$

$$\hat{h}_{max} \geq \hat{h}_j; \quad \forall(j) \tag{41}$$

$$\hat{h}_{total} = \sum_{j=1}^{J} \hat{h}_j \tag{42}$$

$$Z_6 = \hat{h}_{max} \tag{43}$$

$$Z_7 = \hat{h}_{total} \tag{44}$$

3.2.7. Maximum Workload, Total Workload and Maximum Workload-Difference ($Z_8$, $Z_9$ and $Z_{10}$)

In addition to makespan and flowtime, two other objectives commonly considered in FJSP scheduling are the minimization of maximum and total machine workload. They represent the intention of protecting machines from overuse [43]. Moreover, in this research, we noted that, in the presence of alternative routing and sequence-dependent setup time, these objectives could result in a substantially reduced overall system workload with a moderate increase in makespan. This can significantly free up machine operators for other activities, such as quality improvement, development, and maintenance. The necessary variables and constraints to impose these categories of objective functions are discussed below.

The workload on machine $m$ (i.e., $l_{m,o,s,j}$) because of an assigned operation $o$ of sublot $s$ of job $j$ comprises the setup and the actual processing of the operation. The value of this variable is assigned by the constraints in Equations (45)–(48). The overall workload on machine $m$, ($\hat{l}_m$), comprises the workloads because of all the operations assigned to it from the current schedule and its release date $D_m$ as shown in Equation (49). The release date may represent the amount of work that spills into the current planning and scheduling period from the previous one.

The objective function along with the constraints in Equations (50) and (51) set the values of $\hat{l}_{max} = \max_{\forall m}\{l_m\}$ and $\hat{l}_{min} = \min_{\forall m}\{l_m\}$, respectively. The workload difference between the maximally and the least loaded machines (maximum workload difference, $\hat{l}_{diff}$) is calculated using the constraint in Equation (52). The total workload on the system $\hat{l}_{total}$ is evaluated by the constraint in Equation (53). The objective function terms $Z_8$, $Z_9$, and $Z_{10}$ represent the values of $\hat{l}_{max}$, $\hat{l}_{total}$, and $\hat{l}_{diff}$ as enforced by the constraints in Equations (54)–(56), respectively.

$$l_{m,o,s,j} \geq S^*_{o,j,m} + b_{s,j} \cdot T_{o,j,m} - \Omega \cdot (1 - x_{1,m,o,s,j}); \quad \forall(m,o,s,j) \tag{45}$$

$$l_{m,o,s,j} \leq S^*_{o,j,m} + b_{s,j} \cdot T_{o,j,m} + \Omega \cdot (1 - x_{1,m,o,s,j}); \quad \forall(m,o,s,j) \tag{46}$$

$$l_{m,o,s,j} \geq S_{o,j,m,o',j'} + b_{s,j} \cdot T_{o,j,m} - 2\Omega \cdot (1 - x_{r,m,o,s,j} - y_{r-1,m,o',j'}); \quad \forall(r,m,o,s,j)|r>1 \tag{47}$$

$$l_{m,o,s,j} \leq S_{o,j,m,o',j'} + b_{s,j} \cdot T_{o,j,m} + 2\Omega \cdot (1 - x_{r,m,o,s,j} - y_{r-1,m,o',j'}); \quad \forall(r,m,o,s,j)|r>1 \tag{48}$$

$$\hat{l}_m = D_m + \sum_{j=1}^{J} \sum_{s=1}^{S_j} \sum_{o=1}^{O_j} l_{m,o,s,j}; \quad \forall(m) \tag{49}$$

$$\hat{l}_{max} \geq l_m; \quad \forall(m) \tag{50}$$

$$\hat{l}_{min} \leq l_m; \quad \forall(m) \tag{51}$$

$$\hat{l}_{diff} = l_{max} - l_{min}; \quad \forall (m) \tag{52}$$

$$\hat{l}_{total} = \sum_{m=1}^{M} l_m; \tag{53}$$

$$Z_8 = \hat{l}_{max} \tag{54}$$

$$Z_9 = \hat{l}_{total} \tag{55}$$

$$Z_{10} = \hat{l}_{diff} \tag{56}$$

## 4. Genetic Algorithm

### 4.1. Prototype Problem

To illustrate the solution representation and the various genetic operators, a prototype problem that consists of the processing of four jobs using five machines is considered. The complete data sets for this small problem are given in Tables 1 and 2. Data related to batch size ($B_j$), the nature of setup being attached or detached ($A_{o,j}$), lag-time ($L_{o,j}$) and alternative routing ($m$, $T_{o,j,m}$) for each operation are in Table 1. Sequence-dependent setup time data is provided in Table 2. This problem is also used in the numerical example to illustrate the various objective function terms of the proposed model.

**Table 1.** Data for jobs for Problem-1.

| | | | | | | (Eligible Machine, Processing Time) = ($m$, $T_{o,j,m}$) | | | |
|---|---|---|---|---|---|---|---|---|---|
| $j$ | $B_j$ | $S_j$ | $o$ | $A_{o,j}$ | $L_{o,j}$ | i | ii | iii | iv |
| 1 | 100 | 2 | 1 | 0 | 0 | (1, 6.75) | (4, 6.50) | (5, 6.50) | |
| | | | 2 | 1 | 120 | (1, 3.00) | (2, 2.25) | (4, 2.75) | |
| | | | 3 | 0 | 120 | (1, 3.50) | (2, 3.25) | (4, 3.75) | (5, 3.50) |
| 2 | 250 | 3 | 1 | 0 | 0 | (1, 1.75) | (2, 2.00) | (5, 1.25) | |
| | | | 2 | 1 | 0 | (2, 5.00) | (3, 4.25) | (4, 5.00) | (5, 4.75) |
| | | | 3 | 1 | 40 | (1, 7.00) | (2, 7.00) | (3, 6.50) | (5, 6.50) |
| | | | 4 | 0 | 40 | (1, 2.50) | (2, 2.50) | (3, 2.75) | (4, 2.75) |
| 3 | 200 | 3 | 1 | 0 | 0 | (1, 5.25) | (5, 5.75) | | |
| | | | 2 | 1 | 0 | (1, 4.50) | (3, 4.25) | (5, 4.25) | |
| | | | 3 | 1 | 0 | (1, 3.50) | (2, 3.50) | | |
| 4 | 100 | 2 | 1 | 0 | 0 | (4, 6.00) | (5, 6.00) | | |
| | | | 2 | 0 | 0 | (1, 4.25) | (3, 4.75) | (4, 4.75) | (5, 4.75) |
| | | | 3 | 1 | 0 | (2, 2.00) | (4, 1.25) | (5, 1.25) | |

Machine release dates in minutes: $D_1 = 840$, $D_2 = D_3 = 0$, $D_4 = 120$.

**Table 2.** Sequence-dependent setup time data for problem-1.

| | | | Setup Time $S^*_{o,j,m}$ and $S_{o,j,m,o',j'}$ | |
|---|---|---|---|---|
| $j$ | $o$ | $m$ | ($S^*_{o,j,m}$) | $\cdots (j', o', S_{o,j,m,o',j'}) \cdots$ |
| 1 | 1 | 1 | (120) | (1,1,20)(1,2,100)(1,3,120)(2,1,210)(2,3,210)(2,4,240)(3,1,240)(3,2,210)(3,3,240)(4,2,210) |
| | | 4 | (140) | (1,1,15)(1,2,80)(1,3,120)(2,2,180)(2,4,240)(4,1,210)(4,2,210)(4,3,240) |
| | | 5 | (100) | (1,1,20)(1,3,80)(2,1,210)(2,2,180)(2,3,240)(3,1,180)(3,2,240)(4,1,180)(4,2,180)(4,3,180) |
| | 2 | 1 | (140) | (1,1,100)(1,2,20)(1,3,80)(2,1,240)(2,3,210)(2,4,180)(3,1,240)(3,2,210)(3,3,240)(4,2,210) |
| | | 2 | (100) | (1,2,15)(1,3,100)(2,1,180)(2,2,210)(2,3,180)(2,4,180)(3,3,180)(4,3,210) |
| | | 4 | (140) | (1,1,80)(1,2,10)(1,3,80)(2,2,240)(2,4,180)(4,1,240)(4,2,210)(4,3,240) |

**Table 2.** *Cont.*

| j | o | m | $(S^*_{o,j,m})$ | $\cdots (j', o', S_{o,j,m,o',j'}) \cdots$ |
|---|---|---|---|---|
| | 3 | 1 | (80) | (1,1,80)(1,2,120)(1,3,10)(2,1,180)(2,3,240)(2,4,180)(3,1,240)(3,2,180)(3,3,210)(4,2,180) |
| | | 2 | (160) | (1,2,120)(1,3,20)(2,1,180)(2,2,210)(2,3,180)(2,4,180)(3,3,210)(4,3,210) |
| | | 4 | (80) | (1,1,100)(1,2,100)(1,3,15)(2,2,240)(2,4,240)(4,1,240)(4,2,240)(4,3,210) |
| | | 5 | (120) | (1,1,100)(1,3,20)(2,1,180)(2,2,180)(2,3,210)(3,1,180)(3,2,210)(4,1,180)(4,2,210)(4,3,240) |
| 2 | 1 | 1 | (80) | (1,1,240)(1,2,240)(1,3,240)(2,1,20)(2,3,120)(2,4,120)(3,1,210)(3,2,210)(3,3,180)(4,2,240) |
| | | 2 | (120) | (1,2,180)(1,3,180)(2,1,10)(2,2,100)(2,3,120)(2,4,120)(3,3,180)(4,3,180) |
| | | 5 | (160) | (1,1,240)(1,3,180)(2,1,15)(2,2,100)(2,3,100)(3,1,180)(3,2,210)(4,1,180)(4,2,180)(4,3,240) |
| | 2 | 2 | (120) | (1,2,180)(1,3,210)(2,1,80)(2,2,15)(2,3,120)(2,4,100)(3,3,240)(4,3,210) |
| | | 3 | (100) | (2,2,20)(2,3,80)(2,4,120)(3,2,240)(4,2,210) |
| | | 4 | (120) | (1,1,210)(1,2,180)(1,3,240)(2,2,10)(2,4,80)(4,1,180)(4,2,240)(4,3,240) |
| | | 5 | (120) | (1,1,180)(1,3,240)(2,1,120)(2,2,20)(2,3,80)(3,1,210)(3,2,240)(4,1,240)(4,2,210)(4,3,240) |
| | 3 | 1 | (80) | (1,1,210)(1,2,240)(1,3,240)(2,1,80)(2,3,10)(2,4,80)(3,1,240)(3,2,240)(3,3,240)(4,2,210) |
| | | 2 | (100) | (1,2,210)(1,3,180)(2,1,120)(2,2,80)(2,3,15)(2,4,80)(3,3,210)(4,3,180) |
| | | 3 | (140) | (2,2,80)(2,3,10)(2,4,100)(3,2,240)(4,2,180) |
| | | 5 | (160) | (1,1,240)(1,3,210)(2,1,80)(2,2,120)(2,3,10)(3,1,240)(3,2,180)(4,1,180)(4,2,240)(4,3,210) |
| | 4 | 1 | (160) | (1,1,180)(1,2,210)(1,3,210)(2,1,80)(2,3,100)(2,4,10)(3,1,240)(3,2,180)(3,3,180)(4,2,210) |
| | | 2 | (140) | (1,2,180)(1,3,180)(2,1,120)(2,2,80)(2,3,120)(2,4,10)(3,3,180)(4,3,210) |
| | | 3 | (160) | (2,2,100)(2,3,100)(2,4,10)(3,2,240)(4,2,210) |
| | | 4 | (120) | (1,1,240)(1,2,240)(1,3,180)(2,2,120)(2,4,10)(4,1,210)(4,2,180)(4,3,210) |
| 3 | 1 | 1 | (80) | (1,1,240)(1,2,240)(1,3,240)(2,1,180)(2,3,180)(2,4,210)(3,1,15)(3,2,120)(3,3,100)(4,2,180) |
| | | 5 | (80) | (1,1,210)(1,3,240)(2,1,240)(2,2,180)(2,3,240)(3,1,15)(3,2,100)(4,1,210)(4,2,180)(4,3,210) |
| | 2 | 1 | (120) | (1,1,240)(1,2,240)(1,3,180)(2,1,240)(2,3,240)(2,4,240)(3,1,80)(3,2,20)(3,3,100)(4,2,210) |
| | | 3 | (140) | (2,2,240)(2,3,240)(2,4,240)(3,2,15)(4,2,180) |
| | | 5 | (160) | (1,1,180)(1,3,210)(2,1,240)(2,2,180)(2,3,180)(3,1,100)(3,2,10)(4,1,240)(4,2,240)(4,3,240) |
| | 3 | 1 | (120) | (1,1,180)(1,2,240)(1,3,240)(2,1,210)(2,3,180)(2,4,180)(3,1,120)(3,2,100)(3,3,10)(4,2,210) |
| | | 2 | (160) | (1,2,240)(1,3,210)(2,1,210)(2,2,180)(2,3,240)(2,4,180)(3,3,15)(4,3,210) |
| 4 | 1 | 4 | (100) | (1,1,210)(1,2,240)(1,3,210)(2,2,240)(2,4,180)(4,1,10)(4,2,100)(4,3,100) |
| | | 5 | (100) | (1,1,180)(1,3,210)(2,1,180)(2,2,180)(2,3,210)(3,1,210)(3,2,210)(4,1,10)(4,2,120)(4,3,100) |
| | 2 | 1 | (100) | (1,1,240)(1,2,240)(1,3,210)(2,1,180)(2,3,180)(2,4,240)(3,1,180)(3,2,240)(3,3,240)(4,2,20) |
| | | 3 | (80) | (2,2,240)(2,3,240)(2,4,240)(3,2,210)(4,2,20) |
| | | 4 | (140) | (1,1,180)(1,2,210)(1,3,180)(2,2,240)(2,4,240)(4,1,100)(4,2,10)(4,3,100) |
| | | 5 | (80) | (1,1,180)(1,3,180)(2,1,240)(2,2,210)(2,3,210)(3,1,240)(3,2,180)(4,1,80)(4,2,15)(4,3,80) |
| | 3 | 2 | (100) | (1,2,210)(1,3,210)(2,1,210)(2,2,210)(2,3,180)(2,4,210)(3,3,210)(4,3,20) |
| | | 4 | (140) | (1,1,240)(1,2,210)(1,3,240)(2,2,180)(2,4,180)(4,1,120)(4,2,100)(4,3,15) |
| | | 5 | (140) | (1,1,180)(1,3,240)(2,1,210)(2,2,180)(2,3,210)(3,1,240)(3,2,240)(4,1,100)(4,2,100)(4,3,10) |

### 4.2. Solution Encoding

A solution encoding is a technique of transforming a problem statement into a searchable space of all feasible solutions, in which an algorithm can be applied to explore iteratively for optimal solutions. Hence, its design is the first most crucial step in solving a problem using a search-based algorithm. The solution encoding used in this paper combines features from the solution representations in [2] for FJSP lot streaming and that in [1] for dividing the genetic search into two stages. This solution encoding is depicted in Figure 1 for a typical solution of the prototype problem presented in the previous section.

As shown in Figure 1a, the solution representation has two segments. The first segment (Segment-1), detailed in Figure 1b, encodes the numbers and sizes of sublots for all the jobs. The number of genes in this segment is equal to the sum of the maximum number of sublots of each job ($\sum_{j=1}^{J} S_j$), where there are $S_j$ genes corresponding to each job. The

gene $\alpha_{j,s}$ takes a continuous value from the interval [0, 1]. The decoding procedure for the number and sizes of the sublots from Segment-1 is detailed in Section 4.3.1.

| Segment-1 | Segment-2 |
|---|---|
| Encodes the numbers and sizes of the sublots | Encodes the assignment and sequencing of the operations |

(**a**) General structure of the solution representation.

| Details of Segment-1 (Applicable both in Stage-1 and Stage-2 of the GA) | | | | | | | | | |
|---|---|---|---|---|---|---|---|---|---|
| Job-1 | | Job-2 | | | Job-3 | | | Job-4 | |
| $\alpha_{1,1}$ | $\alpha_{1,2}$ | $\alpha_{2,1}$ | $\alpha_{2,2}$ | $\alpha_{2,3}$ | $\alpha_{3,1}$ | $\alpha_{3,2}$ | $\alpha_{3,3}$ | $\alpha_{4,1}$ | $\alpha_{4,2}$ |

(**b**) Details for Segment-1. The gene $\alpha_{j,s}$ takes a continuous value in [0, 1].

| Details of Segment-2 (Applicable only in Stage-1 of the GA) | | | | | | | | | | | | | | | | | | | | | | | | | | | | | | | | |
|---|---|---|---|---|---|---|---|---|---|---|---|---|---|---|---|---|---|---|---|---|---|---|---|---|---|---|---|---|---|---|---|---|
| 1 | 2 | 3 | 4 | 5 | 6 | 7 | 8 | 9 | 10 | 11 | 12 | 13 | 14 | 15 | 16 | 17 | 18 | 19 | 20 | 21 | 22 | 23 | 24 | 25 | 26 | 27 | 28 | 29 | 30 | 31 | 32 | 33 |
| 3,1,1 | 4,1,1 | 1,2,1 | 1,1,1 | 3,2,1 | 2,2,1 | 2,1,1 | 3,3,1 | 4,2,1 | 2,1,2 | 2,1,3 | 3,1,2 | 4,1,2 | 3,1,3 | 2,2,2 | 2,2,3 | 2,2,4 | 2,3,1 | 4,1,3 | 3,2,2 | 2,2,2 | 4,2,3 | 3,2,3 | 4,2,3 | 1,1,2 | 1,1,3 | 2,3,2 | 3,3,2 | 3,3,3 | 1,2,2 | 2,3,3 | 1,2,3 | 2,3,4 |

(**c**) Details for Segment-2 in Stage-1, where a gene takes a value $[j, s, o]$.

| Details of Segment-2 (Applicable only in Stage-2 of the GA) | | | | | | | | | | | | | | | | | | | | | | | | | | | | | | | | |
|---|---|---|---|---|---|---|---|---|---|---|---|---|---|---|---|---|---|---|---|---|---|---|---|---|---|---|---|---|---|---|---|---|
| 1 | 2 | 3 | 4 | 5 | 6 | 7 | 8 | 9 | 10 | 11 | 12 | 13 | 14 | 15 | 16 | 17 | 18 | 19 | 20 | 21 | 22 | 23 | 24 | 25 | 26 | 27 | 28 | 29 | 30 | 31 | 32 | 33 |
| 3,1,1,5 | 4,1,1,5 | 1,2,1,4 | 1,1,1,1 | 3,2,1,5 | 2,2,1,1 | 2,1,1,1 | 3,3,1,1 | 4,2,1,5 | 2,1,2,4 | 2,1,3,3 | 3,1,2,4 | 4,1,2,4 | 3,1,3,1 | 2,2,2,3 | 2,2,3,1 | 2,2,4,3 | 2,3,1,1 | 4,1,3,5 | 3,2,2,5 | 2,2,2,5 | 4,2,2,5 | 3,2,3,5 | 4,2,3,4 | 1,1,2,3 | 1,1,3,1 | 2,3,2,5 | 3,3,2,3 | 3,3,3,2 | 1,2,2,2 | 2,3,3,1 | 1,2,3,5 | 2,3,4,2 |

(**d**) Details for Segment-2 in Stage-2, where a gene takes a value $[j, s, o, m]$.

**Figure 1.** Solution representation.

The second segment (Segment-2) of the solution encoding has two forms. The first form, detailed in Figure 1c, is applicable for the first stage of the search by the genetic algorithm. The number of genes in this segment is equal to the total number of operations in all the sublots, which can be computed as $\sum_{j=1}^{J} \sum_{o=1}^{O_j} S_j \times O_j$. Each gene is a 3-tuple $[j, s, o]$ composed of job, sublot, and operation indices. For a particular $[j, s]$, there are $O_j$ number of genes corresponding to each operation of the sublot, and a gene $[j, s, o]$ appears in the segment earlier than $[j, s, o']$ if $o < o'$.

This segment provides the order (left to right) in which the operations are considered for assignment and sequencing. Whenever an operation of a sublot of a given job is to be assigned to a machine, the algorithm chooses the machine that completes the operation sooner after completing the operations previously assigned to this machine. In that case, the order in which the operations are assigned to machines represents their processing sequence.

The second form of Segment-2, detailed in Figure 1d, is for the second stage of the genetic search. This form of the segment explicitly encodes both the assignment and sequencing of the operations on the machines. Each gene, in this form, is 4-tuple $[j, s, o, m]$ where $m$ encodes the machine assignment for operation $[j, s, o]$ and it is restricted to take the value such that $P_{j,o,m} = 1$. Moreover, for a given $[j, s]$, the gene $[j, s, o, m]$ appeared earlier the sequence than $[j, s, o', m']$ if $o < o'$. The sequence of the operation on a given machine $m$ is dictated by the order in which the genes appeared on Segment-2. For instance, the assignment and the sequence of the operation on machine $m = 4$ is $(j1, s2, o1) \rightarrow (j2, s1, o2) \rightarrow (j4, s1, o2) \rightarrow (j1, s1, o2) \rightarrow (j2, s1, o4)$. The detail discussion of the decoding of Segment-2 under the first and the second stage of the genetic search is given in Section 4.3.2.

### 4.3. Solution Decoding

#### 4.3.1. Number and Size of Sublots

The decoding of the number and sizes of sublots from Segment-1 is similar to that discussed in [2]. Given the values of the genes in Segment-1, the size of a sublot $b_{s,j}$ can be computed using Equation (57). Once all the $b_{s,j}$'s are calculated, a sublot whose size is less than a minimum threshold value is set to zero, and the corresponding gene $\alpha_{s,j}$ is also set to zero. Then, the sizes of the other sublots are reevaluated using the same Equation (57). In this decoding, the number of the sublots for a given job is equal to the number of sublots whose sizes are greater than zero.

$$
b_{s,j} = 
\begin{cases}
\dfrac{\alpha_{s,j}}{\sum_{s=1}^{S_j} \alpha_{s,j}} & \text{if } \sum_{s=1}^{S_j} \alpha_{s,j} > 0 \\[3ex]
B_j / S_j & \text{Otherwise}
\end{cases}
\tag{57}
$$

#### 4.3.2. Assignment, Sequencing, and Completion Time

Once the sizes of all the sublots are known (see Section 4.3.1), the assignment, sequencing, and completion times of the operation of each non-zero sublot and other variables are determined using the information obtained from Segment-2 and two decoding procedures outlined in this section. The first decoding procedure is applicable for Stage-1 of the genetic search, while the second is for Stage-2.

Stage-1

In Stage-1 of the genetic search, the assignment and sequencing of the operations and the determination of their starting and finish times are obtained using a procedure that utilizes the information in the first form of Segment-2 of the solution representation (Figure 1c). In describing this procedure, let us first define GeneS2F1[$l$] to denote the content of a gene $[j, s, o]$ in the first form of Segment-2 at location $l$, where $l$ runs from 1 to the total number of genes in this segment. Moreover, let us define $r_m$ as a run counter for machine $m$, which increases by one every time an operation is assigned to the machine.

With this definition, the steps for the determination of the assignment and sequencing of operations in Stage-1 of the search are outlined in Figure 2 along with the procedure described in Figure 3 to evaluate the decision variables $c_{o,s,j,m}$, $l_{m,o,s,j}$, $e_{s,j}$, and $d_{s,j}$. In Step-1, the counters $l$ and $r_m$ are initialized to 1 and 0, respectively. The values of the indices $j$, $s$, and $o$ are obtained from GeneS2F1[$l$] at Step-2. In Step-3, if $b_{s,j}$ is zero, the algorithm moves to Step-9. Otherwise, it advances to Step-4. In these steps, the counter $r_m$ is temporarily increased by 1 corresponding to all the eligible machines for operation $o$ of job $j$. Then, using the procedure outlined in Figure 3 , the variables $c_{o,s,j,m}$, $l_{m,o,s,j}$, $e_{s,j}$, and $d_{s,j}$ are evaluated corresponding to all these eligible machines.

In Step-5, the machine that finishes operation $[o, s, j]$ with the smallest $c_{o,s,j,m}$ is selected, and in Step-6, the operation is assigned to the $r_m^{th}$ run of this machine. In Step-7, the values of the decision variable calculated corresponding to the selected machine are retained as final values. The values of the counter $r_m$, that was temporarily increased in Step-4, are reduced by 1 corresponding to those machines that are not selected to process operation $[o, s, j]$. In Step-9, the algorithm stops if all the operations are assigned, or otherwise, it increases the counter $l$ by one and then returns to Step-2.

---

**Step 1.** Set $l = 1$. Set $r_m = 0; \forall m$.

**Step 2.** Set $(j, s, o) = \text{GeneS2F1}[l]$ of the chromosome in Figure 1c.

**Step 3.** If $b_{s,j} > 0$, go to Step 4; otherwise, go to Step 9.

**Step 4.** Temporarily set $r_m = r_m + 1$ for each eligible machine $m$ of operation $o$ of job $j$ (for each $m$ such that $p_{o,j,m} = 1$). Using the procedure described in Figure 3 , calculate the completion time of operation $o$ of sublot $s$ of job $j$ corresponding to each of the eligible machines $m$.

**Step 5.** Using the results from Step 4, select the machine that can complete the operation sooner.
Say this machine is machine $m^*$.

**Step 6.** Assign operation $o$ of sublot $s$ of job $j$ to the $(r_{m^*})^{th}$ run of machine $m^*$.

**Step 7.** Retain the values of $c_{o,s,j,m}$, $l_{m,o,s,j}$, $e_{s,j}$, and $d_{s,j}$ calculated in Step 4 corresponding to machine $m = m^*$ as the final values of these variables.

**Step 8.** Set $r_m = r_m - 1$ corresponding to all the other machines considered in Step 4 but not selected to processes operation $o$ of job $j$ in Step 5.

**Step 9.** If $l$ is equal to the total number of operations, stop; otherwise, set $l = l + 1$ and go to Step 2.

---

**Figure 2.** A decoding procedure for the solution representation given in Figure 1c for the first stage of the two-stage genetic algorithm.

Stage-2

In Stage-2 of the genetic search, the second form of Segment-2 of the solution representation (Figure 1d) is used. This form of the segment explicitly encodes the assignment and sequencing of the operations as it was discussed in Section 4.2. Unlike the decoding procedure previously discussed for Stage-1, the decoding in Stage-2 does not follow a greedy approach in selecting a machine for an operation assignment as the assignment and sequencing are directly inferred from the solution representation.

The decoding procedure is only for the determination of several continuous variables along with the start and finish times of the operations of all the sublots with non-zero sizes. This decoding procedure is outlined in Figure 4. In this decoding procedure, the notation GeneS2F2[$l$] denotes the content of the gene [$j, s, o, m$] at location $l$ of the second form of Segment-2. The notations $l$ and $r_m$ have the same meaning as when they were used in the previous discussion.

4.3.3. Calculating Objective Function Terms

In the decoding procedures presented in the previous section, the values of $c_{o,s,j}$, $e_{s,j}$, $d_{s,j}$, and $l_{m,o,s,j}$ were determined. Once the values of these variables are known for each sublot with size greater than zero ($b_{s,j} > 0$), the various terms of the objective function can easily be calculated as shown in Table 3.

If operation $o$ of sublot $s$ sublot of job $j$ is to be processed on $r_m^{th}$ run of machine $m$, the values of the variables $c_{o,s,j}$, $e_{s,j}$, $d_{s,j}$, $\hat{e}_j$, $\hat{d}_j$, and $l_{m,o,s,j}$ are calculated based on one of the following four cases:

---

- **Case 1: $[o = 1; r_m = 1]$**

  (a) Operation $o$ of sublot $s$ of job $j$ is the first operation to be assigned on machine $m$ (i.e., $r_m = 1$), and

  (b) $o = 1$.

  $c_{o,s,j} = D_m + S_{o,j,m}^* + b_{s,j} \times T_{o,j,m}$;
  $l_{m,o,s,j} = S_{o,j,m}^* + b_{s,j} \times T_{o,j,m}$;
  $e_{s,j} = c_{o,s,j,m} - b_{s,j} \times T_{o,j,m} - A_{o,j} \times S_{o,j,m}^*$.

- **Case 2: $[o > 1; r_m = 1]$**

  (a) Operation $o$ of sublot $s$ of job $j$ is the first operation to be assigned on machine $m$ (i.e., $r_m = 1$),

  (b) $o > 1$, and

  (c) Operation $o - 1$ of sublot $s$ of job $j$ was assigned on machine $m'$.

  $c_{o,s,j} = \max\{D_m + (1 - A_{o,j}) \times S_{o,j,m}^* \; , \; c_{o-1,s,j,m'} + L_{o,j}\} + b_{s,j} \times T_{o,j,m} + A_{o,j} \times S_{o,j,m}^*$;
  $l_{m,o,s,j} = S_{o,j,m}^* + b_{s,j} \times T_{o,j,m}$;
  If $o = O_j$, then $d_{s,j} = c_{o,s,j,m}$.

- **Case 3: $[o = 1; r_m > 1]$**

  (a) Operation $o$ of sublot $s$ of job $j$ is not the first operation to be assigned on machine $m$ (i.e., $r_m > 1$),

  (b) Operation $o'$ of sublot $s'$ of job $j'$ is the operation to be processed immediately before operation $o$ of sublot $s$ of job $j$ on machine $m$ (i.e., Operation $o'$ of sublot $s'$ of job $j'$ was assigned to run $r_m - 1$ of machine $m$), and

  (c) $o = 1$.

  $c_{o,s,j} = c_{o',s',j',m} + S_{o,j,m,o',j'} + b_{s,j} \times T_{o,j,m}$;
  $l_{m,o,s,j} = S_{o,j,m,o',j'} + b_{s,j} \times T_{o,j,m}$;
  $e_{s,j} = c_{o,s,j,m} - b_{s,j} \times T_{o,j,m} - A_{o,j} \times S_{o,j,m,o',j'}$.

- **Case 4: $[o > 1; r_m > 1]$**

  (a) Operation $o$ of sublot $s$ of job $j$ is not the first operation to be assigned on machine $m$ (i.e., $r_m > 1$),

  (b) Operation $o'$ of sublot $s'$ of job $j'$ is assigned immediately before operation $o$ of sublot $s$ of job $j$ on machine $m$ (i.e., Operation $o'$ of sublot $s'$ of job $j'$ was assigned to run $r_m - 1$ of machine $m$),

  (c) $o > 1$, and

  (d) Operation $o - 1$ of sublot $s$ of job $j$ is assigned on machine $m'$.

  $c_{o,s,j} = \max\{c_{o',s',j',m} + (1 - A_{o,j}) \times S_{o,j,m,o',j'} \; , \; c_{o-1,s,j,m'} + L_{o,j}\} + b_{s,j} \times T_{o,j,m} + A_{o,j} \times S_{o,j,m,o',j'}$;
  $l_{m,o,s,j} = S_{o,j,m,o',j'} + b_{s,j} \times T_{o,j,m}$;
  If $o = O_j$, then $d_{s,j} = c_{o,s,j,m}$.

**Figure 3.** Calculation of the decision variables $c_{o,s,j}$, $e_{s,j}$, $d_{s,j}$, and $l_{m,o,s,j}$.

**Step 1.** Set $l = 1$. Set $r_m = 0; \forall m$.

**Step 2.** Set $(j, s, o, m) = \text{GeneS2F2}[l]$ of the chromosome in Figure 1d.

**Step 3.** If $b_{s,j} > 0$, go to Step 4; otherwise, go to Step 7.

**Step 4.** Set $r_m = r_m + 1$.

**Step 5.** Assign operation $o$ of sublot $s$ of job $j$ to the $(r_m)^{th}$ run of machine $m$.

**Step 6.** Calculate the values of $c_{o,s,j,m}$, $l_{m,o,s,j}$, $e_{s,j}$, and $d_{s,j}$ using the procedure described in Figure 3.

**Step 7.** If $l$ is equal to the total number of operations, stop; otherwise, set $l = l + 1$ and go to Step 2

**Figure 4.** A decoding procedure for the solution representation given in Figure 1d for the second stage of the two-stage genetic algorithm.

**Table 3.** Calculating the objective function terms once the values of $c_{o,s,j}$, $e_{s,j}$, $d_{s,j}$, and $l_{m,o,s,j}$ are evaluated corresponding to each sublot with non-zero size.

| Intermediate Calculation | Obj. Function Term |
| --- | --- |
| $c_{max} = \max_{\forall (o,s,j)} c_{o,s,j}$ | $Z_1 = c_{max}$ |
| $f_{s,j} = d_{s,j} - e_{s,j}$<br>$f_{max} = \max_{\forall (s,j)\|b_{s,j}>0} \{f_{s,j}\}$ | $Z_2 = f_{max}$ |
| $f_{total} = \sum_{\forall (s,j)\|b_{s,j}>0} \{f_{s,j}\}$ | $Z_3 = f_{total}$ |
| $\hat{e}_j = \min_{\forall s\|b_{s,j}>0} \{e_{s,j}\}$<br>$\hat{d}_j = \max_{\forall s\|b_{s,j}>0} \{d_{s,j}\}$<br>$\hat{f}_j = \hat{d}_j - \hat{e}_j$<br>$\hat{f}_{max} = \max_{\forall j} \{\hat{f}_j\}$ | $Z_4 = \hat{f}_{max}$ |
| $\hat{f}_{total} = \sum_{\forall j} \{\hat{f}_j\}$ | $Z_5 = \hat{f}_{total}$ |
| $\hat{g}_j = \min_{\forall s\|b_{s,j}>0} \{d_{s,j}\}$<br>$\hat{h}_j = \hat{d}_j - \hat{g}_j$<br>$\hat{h}_{max} = \max_{\forall j} \{\hat{h}_j\}$ | $Z_6 = \hat{h}_{max}$ |
| $\hat{h}_{total} = \sum_{\forall j} \{\hat{h}_j\}$ | $Z_7 = \hat{h}_{total}$ |
| $\hat{l}_m = D_m + \sum_{\forall (m,o,s,j)\|b_{s,j}>0} \{l_{m,o,s,j}\}$<br>$\hat{l}_{max} = \max_{\forall m} \{l_m\}$ | $Z_8 = \hat{l}_{max}$ |
| $\hat{l}_{total} = \sum_{\forall m} \{l_m\}$ | $Z_9 = \hat{l}_{total}$ |
| $\hat{l}_{min} = \min_{\forall m} \{l_m\}$<br>$\hat{l}_{diff} = l_{max} - l_{min}$ | $Z_{10} = \hat{l}_{diff}$ |

### 4.4. Handling Multi-Objectives

In the literature, there are many techniques in handling multi-objective optimization using evolutionary algorithms. However, due to its simplicity and computational efficiency, we choose a weighted sum approach in which multiple objectives are aggregated into a single objective using a weight vector. In the best scenario, the weight vector is assigned by decision-makers who have knowledge regarding the relative importance of the objective functions. However, because of large differences in magnitudes of the objective functions, scaling the objectives is always desirable to obtain solutions consistent with the decision-makers' preferences.

Hence, in the aggregated objective, the *k*th objective function has to be multiplied by the weights $W_k$, reflecting the decision makers' preferences, and $\Psi_k$ for scaling as shown in

Equation (58) . In this research, we adopt a simple objective function scaling mechanism in such a way that, in the initial population of the genetic algorithm, the magnitude of the maximum values of objective function terms $Z_2$ through $Z_{10}$ will have the same values as the maximum value of $Z_1$.

This scaling procedure can be mathematically described as shown in Equation (59) where $Z_k^{Ini-max}$ represents the maximum value of objective $Z_k$ in the initial population. The decision-maker is free to choose any positive value of $W_k$. The problem may be solved multiple times with different sets of $W_k$'s, and the resulting solutions can be presented to the decision-makers for final decision. Nevertheless, scheduling is a day-to-day activity where the decision-makers may already have a preferred set of $W_k$'s from previous experience.

$$Z = \sum_{k=1}^{10} W_k \cdot \Psi_k \cdot Z_k \tag{58}$$

$$\Psi_k = \frac{Z_1^{Ini-max}}{Z_k^{Ini-max}} \tag{59}$$

### 4.5. Genetic Operators

A genetic algorithm works on a population of solutions. The initial population is generated randomly, and the algorithm works iteratively to evolve this population towards promising solutions following the principles of natural evolution. The mechanisms used to achieve this artificial evolutionary process are collectively called genetic operators. These operators are broadly classified into selection, crossover, and mutation. The operators used in the proposed genetic algorithm are discussed below.

#### 4.5.1. Selection Operators

The role of selection operator in a genetic algorithm is to mimic the principle of the survival of the fittest in natural evolution. This operator creates a mating pool of individuals for reproduction. Selection can be applied in a variety of ways. In this research, we considered the three most commonly used approaches in the literature—namely, (1) proportional, (2) linear ranking, and (3) tournament selections. The following notations are used to describe these selection operators.

$N$ — Number of individuals (solutions) in a population.

$U(t)$ — Population of solution at generation $t$.

$U(i, t)$ — The $i$th individual in the population at generation $t$.

$M(t)$ — Mating pool created via selection operator from the population $U(t)$ (the size of the mating pool is the same as that of the population).

$M(i, t)$ — The $i$th individual in the mating pool at generation $t$.

$Z(i, t)$ — The weighted objective function value corresponding to the $i$th individual in the population at generation $t$.

$Z_{min}(t)$ — The minimum observed weighted objective function value in the population at generation $t$.

$Z_{max}(t)$ — The maximum observed weighted objective function value in the population at generation $t$.

$F(i, t)$ — The fitness value of the $i$th individual in the population at generation $t$.

$R(i, t)$ — The rank of the $i$th individual in the population at generation $t$ for linear ranking selection.

$P(i, t)$ — Probability of selection of $i$th individual in the population at generation $t$ for proportional or linear ranking selection method.

$T$ — Tournament size for tournament selection.

### 4.5.2. Proportional Selection

Proportional selection is a procedure in which individuals from a given generation are selected (with replacement) to move to the mating pool with a probability proportional to their fitness $F$, which needs to be maximized. In a problem where the objective function $Z$ is to be minimized, a fitness function $F$ has to be devised so that a solution with smaller $Z$ will have higher fitness than a solution with larger $Z$.

In such situations, a commonly used fitness function is the reciprocal of $Z$ as shown in Equation (60) . We also considered other two transformations shown in Equations (61) and (62). Once the fitness values for all the individuals in the population are calculated, each individual is assigned a probability of selection defined by the equation in Equation (63). As can be seen from this equation, $P(i, t)$ is proportional to the fitness $F(i, t)$, and the sum $\sum_i^N P(i, t)$ is equal to 1. This probability distribution can be sampled using Monte-Carlo simulation of a roulette wheel, where each solution is assigned a slot proportional to its probability of selection ($P(i, t)$). Algorithm 1 depicts the Monte-Carlo simulation of a roulette wheel, and every time this algorithm is called, it returns an integer number (*winner*) representing the index of the selected individual. The procedure of constituting the mating pool $M(t)$ from a given population $U(t)$ is depicted in Algorithm 2.

$$F(i, t) = \frac{1}{Z(i, t)} \tag{60}$$

$$F(i, t) = Z_{max}(t) + Z_{min}(t) - Z(i, t) \tag{61}$$

$$F(i, t) = Z_{max,t} - Z(i, t) \tag{62}$$

$$P(i, t) = \frac{F(i, t)}{\sum_i^N F(i, t)} \tag{63}$$

### 4.5.3. Linear Ranking Selection

In a linear ranking selection, the individuals in the population are assigned ranks based on a sorted sequence of their objective function values. The individuals with the worst objective function are assigned a rank of 1, the next worse individuals are assigned a rank of 2, and so on. In this process, the best individuals are assigned the highest possible rank. Once each individual is a assigned a rank R(i,t), a selection probability $P(i, t)$ can be calculated using Equation (64). This probability function can be sampled using Monte-Carlo simulation of a roulette wheel (Algorithm 1) to constitute the mating pool using Algorithm 2.

$$P(i, t) = \frac{R(i, t)}{\sum_i^N R(i, t)} \tag{64}$$

### 4.5.4. Tournament Selection

Tournament selection is the most commonly used selection operator in the literature. In this selection procedure, every time a selection is performed, $T$ individuals are randomly selected (with replacement) from the population, and the one with the smallest $Z$ is selected as a winner. The process is repeated for $N$ number of times to form a mating pool of $N$ individuals from a given generation of the population. The integer parameter $T$ is referred to as the tournament size, and it is usually equal to a small fraction of $N$ where the smallest possible value is 2. A large value of $T$ results in higher selection pressure and premature convergence, whereas a small value of $T$ may result in slow convergence. The Monte-Carlo

simulation of tournament selection is given in Algorithm 3, which can be used along with Algorithm 2 to constitute the mating pool.

---

**Algorithm 1:** Monte Carlo simulation of roulette wheel spinning for proportional or ranked selection.

**Input** : $P(i, t)$ for $i = 1, 2, ..., N$
**Output**: *Winner*

1 **Set** *Sum* $= 0$
2 **Set** $\rho = \text{rand}()$
   /* **Assign $\rho$ a random number between 0 and 1 using random number generator function rand() */**
3 **for** $i = 1$ **to** $N$ **do**
4     $Sum = Sum + P(i, t)$
5     **if** $\rho \leq Sum$ **then**
6        $Winner = i$
7        **Break**
   /* **Break the "for loop" and go to line 10 */**
8     **end**
9 **end**
10 **Return** *Winner*

---

**Algorithm 2:** Creating the mating pool $M(t)$.

**Input** : $U(t)$
**Output**: $M(t)$

1 **for** $i = 1$ **to** $N$ **do**
2     $j = \text{Selection}()$
   /* **The function Selection() returns the index of the individual selected from $U(t)$. This function is implemented either using Algorithm-1 if roulette wheel selection is used or Algorithm-3 if tournament selection is used */**
3     $M(i, t) = U(j, t)$
4 **end**
5 **Return** *Winner*

---

**Algorithm 3:** Monte Carlo Simulation of Tournament selection.

**Input** : $Z(i, t)$ for $i = 1, 2, ..., N$
**Output**: *Winner*

1 **for** $j = 1$ **to** $T$ **do**
2     $Competitor[j] = \text{RandIntBetween}(1, N)$
   /* **Select $T$ competitors randomly */**
3 **end**
4 $Winner = Competitor[1]$
   /* **Assign Winner the index of the first competitor */**
5 **for** $j = 2$ **to** $T$ **do**
6     $w = Winner$
7     $i = Competitor[j]$
8     **if** $Z(i, t) < Z(w, t)$ **then**
9        $Winner = i$
10     **end**
11 **end**
12 **Return** *Winner*

### 4.5.5. Crossover Operators

Crossover operators are responsible for creating offspring from parent chromosomes via the exchange of genetic materials, thereby, mimicking sexual reproduction in living organisms. Once $M(t)$ is formed using the selection operator, each individual is paired randomly to create a total of $N/2$ pairs. Then, a crossover operator is applied on each pair resulting from the creation of offspring. The crossover operators used in this paper are listed below. SSC1, SSC2, JLOSC, SLOSC, and MAC are direct adaptations from [2]. However, in this paper, JLOSC, SLOSC, and MAC are applicable only in the second stage of the search. JLSCS and SLGSC share similarities with JLOSC and SLOSC; however, they are applicable only in the first stage of the genetic search.

(a)    Sublot-Size Crossover-1 (SSC1).
(b)    Sublot-Size Crossover-2 (SSC2).
(c)    Job Level Gene Sequence Crossover (JLGSC).
(d)    Sublot Level Gene Sequence Crossover (SLGSC).
(e)    Job Level Operation Sequence Crossover (JLOSC).
(f)    Sublot Level Operation Sequence Crossover (SLOSC).
(g)    Machine Assignment Crossover (MAC).

Figure 5 depicts the first two crossover operators (SSC1 and SSC2). When SSC1 (or SSC2) is applied, an arbitrary crossover point is selected on Segment-1, and the parts of this segment that lie to the left (or right) of the crossover point are exchanged. The step-by-step application of JLGSC is illustrated in Figure 6 where the creation of Child-1 is detailed. In Step-1, one gene is selected arbitrarily. This gene and all the other genes with the same job index, $j$, are copied from Parent-1 to Child-1. In Step-3, all the missing genes of Child-1 are copied from parent-2 in the order they appeared in this second parent. At the same time, Child-2 is also created by first copying genes from Parent-2 with the same job index as the arbitrarily selected gene. The missing genes of Child-2 will be obtained from Parent-1. SLGSC is applied in a similar manner to JLGSC. However, the gene transfer in SLGSC is limited to the genes that have the same job and sublot index $(j, s)$ as the arbitrarily selected gene in Step-1.

JLOSC is applied in four steps as shown in Figure 7. In Step-1, a crossover point (a gene) is selected arbitrarily. All the genes with the same job index as the arbitrarily selected gene are copied from Parent-1 to Child-1 in Step-2. In Step-3, the first three elements $(j, s, o)$ of the missing genes of Child-1 are copied from Parent-2. In the last step, the machine assignments of the incomplete genes that were copied from Parent-2 are completed by the machine assignment obtained from Parent-1.

The creation of Child-2 will be performed in a similar manner by starting from Parent-2. SLOSC is a reduced version of JLOSC where the first step is limited to the genes with the same job and sublot indices. Hence, if Figure 7 were for SLOSC, only the genes with job index $j = 3$ and sublot index $s = 2$ would be copied to Child-1 in Step-2. When either JLOSC or SLOSC is applied, Child-1 (Child-2) will have the same machine assigned as Parent-1 (Parent-2) but with a different operation sequence. Thus, JLOSC and SLOSC manipulate only the sequence of operations without altering the machine assignments in Stage-2 of the generic search.

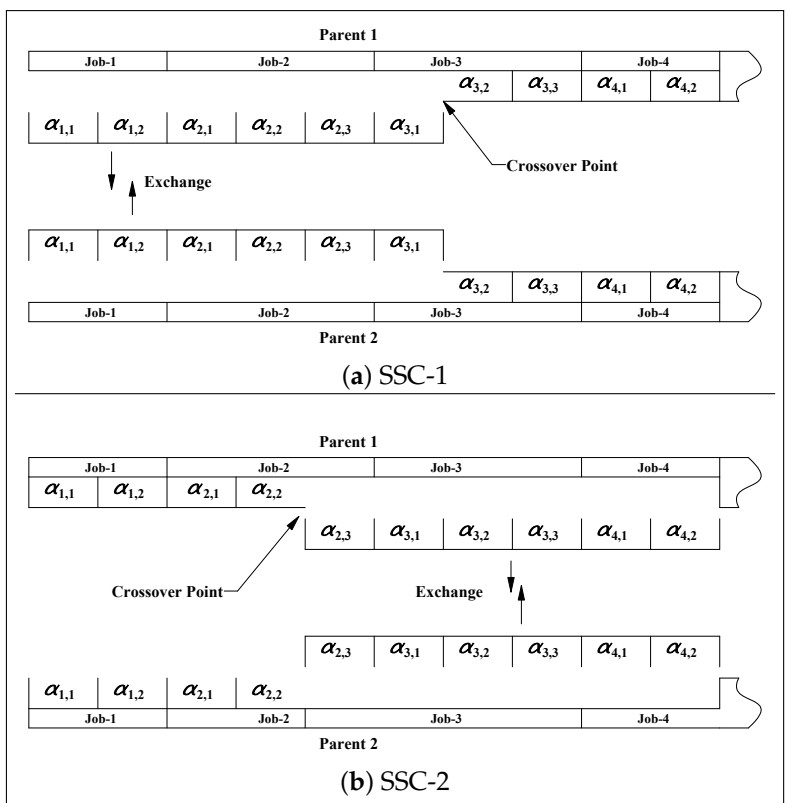

**Figure 5.** Illustration of the crossover operators SSC-1 and SSC-2.

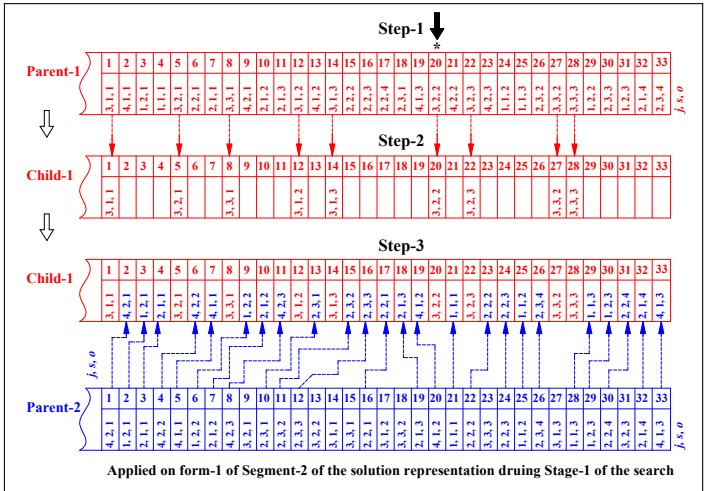

**Figure 6.** Illustration of the JLGSC crossover operator (an asterisk * in Step-1 denotes the location of an arbitrarily selected gene).

The machine assignment crossover (MAC), shown in Figure 8, is responsible for exchanging machine assignment information between parent chromosomes during Stage-2 of the genetic search. As can be seen in the figure, this operator is applied in three steps to create offspring. In Step-1, several genes are arbitrarily selected (each one with 50% chance). In Step-2, the contents of Parent-1 are copied to Child-1 without the machine assignment information of the arbitrarily chosen genes. In the last step, the missing machine assignment information is copied from Parent-2.

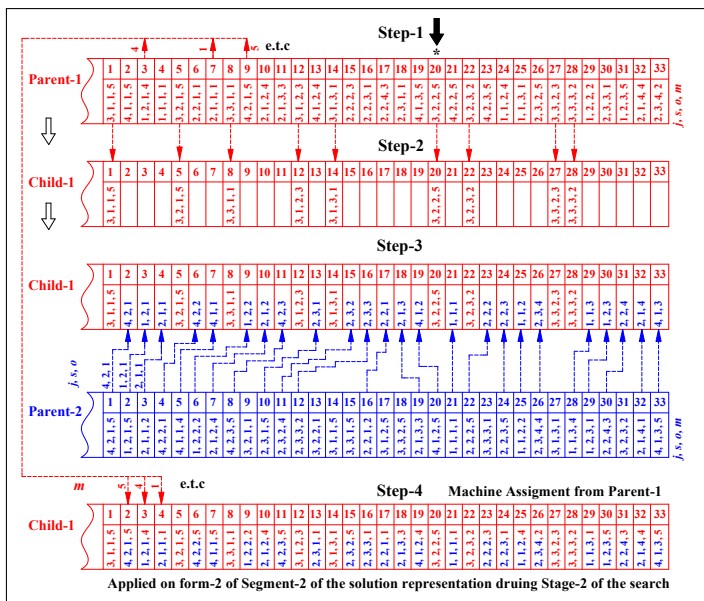

**Figure 7.** Illustration of the JLOSC crossover operator (an asterisk * in Step-1 denotes the location of an arbitrarily selected gene).

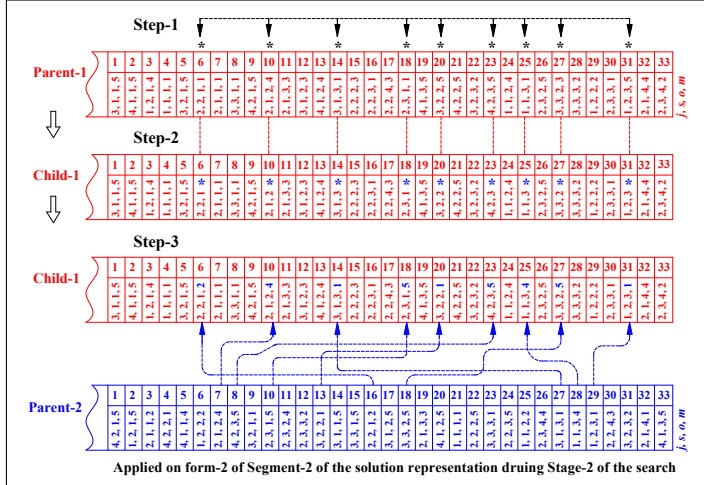

**Figure 8.** Illustration of the MAC crossover operator (the asterisks * denote the locations of arbitrarily selected genes).

Child-2 is created in a similar manner by starting Step-1 from Parent-2 for the same locations of the arbitrarily selected genes in creating Child-1 (i.e., locations 6, 10, 14, 18, 20, 23, 25, 27, and 31). Here, it is essential to mention that, although there are many crossover operators, whenever crossover is to happen between a pair of parent chromosomes, only one crossover operator will be arbitrarily selected and applied with a probability ($p_{cros}$) to create two offspring. If the selected crossover operator is not applied (by chance), the parent chromosomes will move to the next generation (with or without mutation operators applied, again by chance).

### 4.5.6. Mutation Operators

Mutation operators are applied with a small probability on newly generated offspring to alter the genetic material. This category of operators used in this paper is listed below. They are adapted from [2], and the need to divide the genetic search into two stages is taken into consideration. The first two mutation operators, SGVM and SGSM, are applicable in both Stage-1 and -2 of the GA search. Whereas OGSM is applicable only in Stage-1, and

OSSM, ROAM, and IOAM are applicable only in Stage-2. Each one of the six mutation operators listed below is applied with a probability $p_{mut}$ on a newly generated offspring as long as it is eligible for the stage of the search of the GA.

(a) Sublot Gene Value Mutation (SGVM).
(b) Sublot Gene Swap Mutation (SGSM).
(c) Operation Gene Shift Mutation (OGSM).
(d) Operations Sequence Shift Mutation (OSSM).
(e) Random Operation Assignment Mutation (ROAM).
(f) Intelligent Operations Assignment Mutation (IOAM).

SGVM is used to alter the value of a gene $\alpha_{j,s}$ in Segment-1 of a newly born offspring chromosome. When this operator is applied, a single gene is arbitrarily selected, and its value is either increased or decreased (50% by chance) with a small quantity according to Equations (65) or (66), respectively. In these two equations, rand() is a function that returns a random number in the interval [0, 1], and $\delta$ is the maximum increment or decrement quantity, which can be regarded as the GA's parameter that needs to be set. In this research, we found that a value of $\delta$ between 0.1 and 0.2 is preferable. The second mutation operator, SGSM, arbitrarily selects a job $j$ in Segment-1 and swaps the values of two arbitrarily selected genes $\alpha_{j,s}$ and $\alpha_{j,s'}$ corresponding to sublots $s$ and $s'$ ($s \neq s'$).

$$\alpha_{j,s} = \min\{1, \alpha_{j,s} + \text{rand}() \times \delta\} \tag{65}$$

$$\alpha_{j,s} = \max\{1, \alpha_{j,s} - \text{rand}() \times \delta\} \tag{66}$$

Operation gene shift mutation (OGSM) is applied on form-1 of Segment-2 of the solution representation (Figure 1c) during Stage-1 of the GA search. When this operator is applied, it first arbitrarily selects a gene $(j, s, o)$, and then relocates it to an arbitrarily location after and before the locations of the genes $(j, s, o - 1)$ and $(j, s, o + 1)$, respectively. If $o = 0$, the selected gene $(j, s, o)$ can be moved only forward to an arbitral location before the location of gene $(j, s, o + 1)$, whereas if $o = O_j$, the gene can be moved only backward to a location after the location of the gene $(j, s, o - 1)$.

OGSM impacts both the machine assignment and operation sequencing as it alters the sequence of the genes, which is used to determine the operations assignment and sequencing in Stage-1 of the GA search by the greedy procedure described in Section 4.3.2. OSSM is applied on form-2 of Segment-2 (Figure 1d) to shift a location of an arbitrarily selected gene $(j, s, o, m)$ during Stage-2 of the search. It is applied in a similar manner as OGSM was applied during Stage-1. However, in Stage-2, since the genes directly encode the machine assignment, OSSM impacts only the operation sequencing but not the machine assignment of the operations. ROAM is a mutation operator responsible for altering the machine assignment in Stage-2 of the GA search.

The operator arbitrarily selects a gene $(j, s, o, m)$, and changes the value of $m$ to a different eligible machine $m'$ for operation $o$ of job $j$ (i.e., $P_{o,j,m'} = 1$). IOAM intelligently changes a machine assignment in an attempt to lower the workload on a heavily loaded machine. This operator first identifies the machine with the largest workload because of the solution under consideration for a mutation (let that machine be donated as $m^*$). Then, it selects one of the operations assigned to $m^*$ and relocates it to an eligible machine with the least load as long as the load transfer will not make the least loaded machine more loaded than $m^*$ after the load transfer.

## 5. Numerical Studies

### 5.1. Model Analysis

5.1.1. Illustration of Objective Function Terms

This section attempts to illustrate the various objective function terms considered in this paper. For this purpose, the prototype problem presented in Section 4.1 was solved using the proposed algorithm, where makespan minimization was the only objective function, and other objective function terms were merely evaluated. The details of a typical

solution are given in Tables 4–8. Table 4 provides the sizes of the created sublots and operation-to-machine assignments and the run orders along with the Beginning and End times of Lag-time, Setup, and Processing. The maximum Processing End time, 2603.8, is the makespan of the schedule.

**Table 4.** Operation scheduling for Problem-1.

| $j$ | $s$ | $b_{s,j}$ | $o$ | $m$ | $r$ | **LB** | **LE** | **SB** | **SE/PB** | **PE** |
|---|---|---|---|---|---|---|---|---|---|---|
| 1 | 1 | 100.0 | 1 | 5 | 1 | 0.0 | 0.0 | 0.0 | 100.0 | 750.0 |
| | | | 2 | 4 | 6 | 750.0 | 870.0 | 1577.5 | 1817.5 | 2092.5 |
| | | | 3 | 4 | 7 | 2092.5 | 2212.5 | 2112.5 | 2212.5 | 2587.5 |
| 2 | 1 | 90.8 | 1 | 2 | 2 | 0.0 | 0.0 | 303.0 | 313.0 | 494.6 |
| | | | 2 | 3 | 2 | 494.6 | 494.6 | 792.0 | 812.0 | 1197.8 |
| | | | 3 | 3 | 3 | 1197.8 | 1237.8 | 1237.8 | 1317.8 | 1907.8 |
| | | | 4 | 3 | 4 | 1907.8 | 1947.8 | 1907.8 | 2007.8 | 2257.4 |
| | 2 | 67.7 | 1 | 2 | 3 | 0.0 | 0.0 | 494.6 | 504.6 | 640.0 |
| | | | 2 | 2 | 4 | 640.0 | 640.0 | 640.0 | 720.0 | 1058.5 |
| | | | 3 | 2 | 5 | 1058.5 | 1098.5 | 1098.5 | 1178.5 | 1652.5 |
| | | | 4 | 2 | 7 | 1652.5 | 1692.5 | 2308.1 | 2428.1 | 2597.4 |
| | 3 | 91.5 | 1 | 2 | 1 | 0.0 | 0.0 | 0.0 | 120.0 | 303.0 |
| | | | 2 | 3 | 1 | 303.0 | 303.0 | 303.0 | 403.0 | 792.0 |
| | | | 3 | 2 | 6 | 792.0 | 832.0 | 1652.5 | 1667.5 | 2308.1 |
| | | | 4 | 3 | 5 | 2308.1 | 2348.1 | 2338.1 | 2348.1 | 2599.8 |
| 3 | 1 | 80.4 | 1 | 1 | 1 | 0.0 | 0.0 | 840.0 | 920.0 | 1342.0 |
| | | | 2 | 1 | 2 | 1342.0 | 1342.0 | 1342.0 | 1422.0 | 1783.8 |
| | | | 3 | 1 | 3 | 1783.8 | 1783.8 | 1783.8 | 1883.8 | 2165.2 |
| | 2 | 39.2 | 1 | 5 | 2 | 0.0 | 0.0 | 750.0 | 960.0 | 1185.5 |
| | | | 2 | 5 | 5 | 1185.5 | 1185.5 | 2104.4 | 2114.4 | 2281.1 |
| | | | 3 | 1 | 5 | 2281.1 | 2281.1 | 2456.5 | 2466.5 | 2603.8 |
| | 3 | 80.4 | 1 | 5 | 3 | 0.0 | 0.0 | 1185.5 | 1200.5 | 1662.8 |
| | | | 2 | 5 | 4 | 1662.8 | 1662.8 | 1662.8 | 1762.8 | 2104.4 |
| | | | 3 | 1 | 4 | 2104.4 | 2104.4 | 2165.2 | 2175.2 | 2456.5 |
| 4 | 1 | 50.0 | 1 | 4 | 2 | 0.0 | 0.0 | 520.0 | 530.0 | 830.0 |
| | | | 2 | 4 | 3 | 830.0 | 830.0 | 830.0 | 930.0 | 1167.5 |
| | | | 3 | 5 | 6 | 1167.5 | 1167.5 | 2281.1 | 2521.1 | 2583.6 |
| | 2 | 50.0 | 1 | 4 | 1 | 0.0 | 0.0 | 120.0 | 220.0 | 520.0 |
| | | | 2 | 4 | 4 | 520.0 | 520.0 | 1167.5 | 1177.5 | 1415.0 |
| | | | 3 | 4 | 5 | 1415.0 | 1415.0 | 1415.0 | 1515.0 | 1577.5 |

LB = Lag-time Begins; LE = Lag-time Ends; SB = Setup Begins; SE = Setup Ends; PB = Processing Begins; and PE = Processing Ends.

**Table 5.** Sublot flowtime related performance measure ($f_{max}$ and $f_{total}$).

| $j$ | $s$ | $e_{s,j}$ | $d_{s,j}$ | $f_{s,j}$ |
|---|---|---|---|---|
| 1 | 1 | 100.0 | 2587.5 | 2487.5 |
| 2 | 1 | 313.0 | 2257.4 | 1944.4 |
| | 2 | 504.6 | 2597.4 | 2092.8 |
| | 3 | 120.0 | 2599.8 | 2479.8 |
| 3 | 1 | 920.0 | 2165.2 | 1245.2 |
| | 2 | 960.0 | 2603.8 | 1643.8 |
| | 3 | 1200.5 | 2456.5 | 1256.0 |
| 4 | 1 | 530.0 | 2583.6 | 2053.6 |
| | 2 | 220.0 | 1577.5 | 1357.5 |
| | | | Maximum | 2487.5 |
| | | | Total | 16,560.6 |

**Table 6.** Job flowtime and sublot finish-time separation performance measures ($\hat{f}_{mas}$, $\hat{f}_{total}$, $\hat{h}_{mas}$, and $\hat{h}_{total}$).

| $j$ | $\hat{e}_j$ | $\hat{d}_j$ | $\hat{f}_j$ | $\hat{h}_j$ |
|---|---|---|---|---|
| 1 | 100.0 | 2587.5 | 2487.5 | 0.0 |
| 2 | 120.0 | 2599.8 | 2479.8 | 342.4 |
| 3 | 920.0 | 2603.8 | 1683.8 | 438.6 |
| 4 | 220.0 | 2583.6 | 2363.6 | 1006.1 |
| | | Maximum | 2487.5 | 1006.1 |
| | | Total | 9014.7 | 1787 |

The values of the objective function terms related to sublot flowtime can be extracted from Table 4. The entry time to the shop floor of a sublot ($e_{s,j}$) is the setup begin (SB) time of the first operation if the setup is attached, or it is equal to the setup end (SE) time if the setup is detached. Here, it is important to note that, if the setup of the first operation is detached, the setup can begin and be completed before raw material is dispatched to the shop floor. The sublot departure time ($d_{s,j}$) is the process end (PE) time of the last operation.

From the first row of column ten of Table 4, $e_{1,1} = 100$, because the job is dispatched after its setup is completed as the first operation of this sublot has a detached setup time. The departure time $d_{1,1} = 2587.5$. Hence, the flowtime $f_{1,1} = 2587.5 - 100 = 2487.5$. The flowtimes for the other sublots can be determined similarly and are summarized in Table 5. At the bottom of this table, the performance measures $f_{max}$ and $f_{total}$ are indicated as 2487.5 and 16,560.6, respectively.

The job flowtime and sublot-finish-separation objective function terms can be evaluated from Table 5. The entry time $\hat{e}_1$ and the departure time $\hat{d}_1$ of the first job are the same as $e_{1,1}$ and $d_{1,1}$, respectively, of the first sublot since this job has only one sublot in the final solution. Hence, its flowtime $\hat{f}_1 = f_{1,1} = 2487.5$. The second job has three sublots in the final solution. Thus, its entry time $\hat{e}_2$ is the minimum of $\{e_{1,2}, e_{2,2}, e_{3,2}\} = e_{3,2} = 120.0$, and its departure time $\hat{d}_2$ is the maximum of $\{d_{1,2}, d_{2,2}, d_{3,2}\} = d_{3,2} = 2599.8$. Therefore, the flowtime of job-2 is evaluated as $\hat{f}_2 = 2599.8 - 120.0 = 2479.5$. For job-3, $\hat{e}_3$ is the minimum of $\{e_{1,3}, e_{2,3}, e_{3,3}\} = e_{1,3} = 920.0$, and $\hat{d}_3$ is the maximum of $\{d_{1,3}, d_{2,3}, d_{3,3}\} = d_{2,3} = 2603.8$.

Therefore, $\hat{f}_3 = 2603.8 - 920.0 = 1683.8$. Similarly, for the forth job, $\hat{e}_4 = 220.0$, $\hat{d}_4 = 2583.6$, and $\hat{f}_4 = 2363.6$. The sublot finish separation time for job-1 is zero since this job has only one sublot, whereas the sublot finish-time separation of job-2 is the difference between (1) the maximum of $\{d_{1,2}, d_{2,2}, d_{3,2}\} = d_{3,2} = \hat{d}_2$, and (2) the minimum of $\{d_{1,2}, d_{2,2}, d_{3,2}\} = d_{1,2} = \hat{g}_2$, which is evaluated as $\hat{h}_2 = \hat{d}_2 - \hat{g}_2 = 2599.8 - 2257.4 = 342.4$. The sublot finish-time separations for the other jobs can be evaluated similarly. The result is summarized in Table 6. In the last row of this table, the objective function components $\hat{f}_{mas}$, $\hat{f}_{totoal}$, $\hat{h}_{mas}$, and $\hat{h}_{totoal}$ are indicated as 2487.5, 9014.7, 1006.1, and 1787, respectively.

Table 7 provides the schedule for the prototype problem with respect to the machines. From this table, the workload because of each operation assignment can be evaluated by subtracting SB from PE. For instance, from the first row of this table, the load because of operation-1 of sublot-1 of job-3 is $l_{m,o,s,j} = l_{1,1,1,3} = 1342.0 - 840.0 = 502$. Similarly, the workload because of the other four operations on machine-1 can be evaluated as 441.8, 381.4, 291.3, and 147.3, bringing the total workload on this machine to $1763.8 + 840 = 2603.8$, where 840 is the release date of the machine.

The utilization (workload/makespan) of this machine is 100% as its workload is the same as the makespan. The workloads and the utilization of the other machines are also evaluated in a similar way as summarized in Table 8. The objective function components $\hat{l}_{max}$, $\hat{l}_{total}$, and $\hat{l}_{max} - \hat{l}_{min}$ are given in the last row of this table as 2603.8, 12,488.4, and 427.7, respectively. The values for all of the objective function components are summarized in Table 9.

**Table 7.** Operation scheduling for Problem-1 with reference to the machines.

| *m* | *r* | *j* | *s* | *o* | **SB** | **SE/PB** | **PE** |
|---|---|---|---|---|---|---|---|
| 1 | 1 | 3 | 1 | 1 | 840.0 | 920.0 | 1342.0 |
| | 2 | 3 | 1 | 2 | 1342.0 | 1422.0 | 1783.8 |
| | 3 | 3 | 1 | 3 | 1783.8 | 1883.8 | 2165.2 |
| | 4 | 3 | 3 | 3 | 2165.2 | 2175.2 | 2456.5 |
| | 5 | 3 | 2 | 3 | 2456.5 | 2466.5 | 2603.8 |
| 2 | 1 | 2 | 3 | 1 | 0.0 | 120.0 | 303.0 |
| | 2 | 2 | 1 | 1 | 303.0 | 313.0 | 494.6 |
| | 3 | 2 | 2 | 1 | 494.6 | 504.6 | 640.0 |
| | 4 | 2 | 2 | 2 | 640.0 | 720.0 | 1058.5 |
| | 5 | 2 | 2 | 3 | 1098.5 | 1178.5 | 1652.5 |
| | 6 | 2 | 3 | 3 | 1652.5 | 1667.5 | 2308.1 |
| | 7 | 2 | 2 | 4 | 2308.1 | 2428.1 | 2597.4 |
| 3 | 1 | 2 | 3 | 2 | 303.0 | 403.0 | 792.0 |
| | 2 | 2 | 1 | 2 | 792.0 | 812.0 | 1197.8 |
| | 3 | 2 | 1 | 3 | 1237.8 | 1317.8 | 1907.8 |
| | 4 | 2 | 1 | 4 | 1907.8 | 2007.8 | 2257.4 |
| | 5 | 2 | 3 | 4 | 2338.1 | 2348.1 | 2599.8 |
| 4 | 1 | 4 | 2 | 1 | 120.0 | 220.0 | 520.0 |
| | 2 | 4 | 1 | 1 | 520.0 | 530.0 | 830.0 |
| | 3 | 4 | 1 | 2 | 830.0 | 930.0 | 1167.5 |
| | 4 | 4 | 2 | 2 | 1167.5 | 1177.5 | 1415.0 |
| | 5 | 4 | 2 | 3 | 1415.0 | 1515.0 | 1577.5 |
| | 6 | 1 | 1 | 2 | 1577.5 | 1817.5 | 2092.5 |
| | 7 | 1 | 1 | 3 | 2112.5 | 2212.5 | 2587.5 |
| 5 | 1 | 1 | 1 | 1 | 0.0 | 100.0 | 750.0 |
| | 2 | 3 | 2 | 1 | 750.0 | 960.0 | 1185.5 |
| | 3 | 3 | 3 | 1 | 1185.5 | 1200.5 | 1662.8 |
| | 4 | 3 | 3 | 2 | 1662.8 | 1762.8 | 2104.4 |
| | 5 | 3 | 2 | 2 | 2104.4 | 2114.4 | 2281.1 |
| | 6 | 4 | 1 | 3 | 2281.1 | 2521.1 | 2583.6 |

SB = Setup Begins; SE = Setup Ends; PB = Processing Begins; and PE = Processing Ends.

**Table 8.** Machine workload related performance ($\hat{l}_{max}$ and $\hat{l}_{max} - \hat{l}_{min}$).

| *m* | **Workload** | **Utilization** |
|---|---|---|
| 1 | 2603.8 | 100.0 |
| 2 | 2557.4 | 98.2 |
| 3 | 2176.1 | 83.6 |
| 4 | 2567.5 | 98.6 |
| 5 | 2583.6 | 99.2 |

Maximum workload = 2603.8; Total workload = 12488.4; and Maximum workload difference = 427.7.

### 5.1.2. Optimizing a Single Objective

When we optimize only one objective function term, unaccounted objective function terms can be adversely impacted. This phenomenon asserts the importance of multi-objective optimization. To illustrate this reality, we solve Problem-1 by considering one objective function term at a time. The result is depicted in Figure 9. Figure 9a provides the values of the makespan ($Z_1$) when $Z_1$ through $Z_{10}$ are optimized one at a time as a single objective function. As it should be the case, the smallest makespan (about 2608) is achieved when $Z_1$ is considered as the only term in the objective function.

**Table 9.** Values of the objective function components.

| Objective Term | Notation | Value |
|---|---|---|
| Makespan | $Z_1$ | 2603.8 |
| Maximum Sublot Flowtime | $Z_2$ | 2487.5 |
| Total Sublot Flowtime | $Z_3$ | 16,560.6 |
| Maximum Job flowtime | $Z_4$ | 2487.5 |
| Total job flowtime | $Z_5$ | 9014.7 |
| Maximum Sublot Finish-time Separation | $Z_6$ | 1006.1 |
| Total Sublot finish-time Separation | $Z_7$ | 1787.1 |
| Maximum Machine Load | $Z_8$ | 2603.8 |
| Total Machine Load | $Z_9$ | 2603.8 |
| Maximum Machine Load Difference | $Z_{10}$ | 427.7 |

However, when another term alone is optimized, makespan greatly deteriorates. For instance, when only $Z_2$ alone is optimized, the value of the makespan increases to 5067 (94% increase). Figures 9b is a plot of the maximum sublot flowtime ($Z_2$) when $Z_1$ through $Z_{10}$ are optimized one at a time. Its minimum value is 1468 when $Z_2$ alone is optimized. This value increases to 2431 when $Z_1$ alone is optimized. A single objective optimization of $Z_6$ through $Z_{10}$ has significantly negative impacts on $Z_2$. A similar phenomenon is observed on all the other objective function terms, as can be seen from Figures 9c–j.

Here, it is important to note that the magnitude of the severity of a single objective optimization on the objective function terms that are not incorporated increases as the problem size increases. To exemplify this fact, we conducted a similar analysis on a relatively large problem (Problem-4), and the result is compiled in Table 10. For instance, when the total machine load ($Z_9$) was the only objective function, its value was 573,164 min (see at row-$Z_9$ column-$Z_9$), which increases by 84,386 min when makespan ($Z_1$) is the only objective function optimized (see at row-$Z_1$ column-$Z_9$). This increment was only 721 min in a similar analysis in Problem-1. The total sublot flow time in Problem-4 was equal to 2,115,095 when it was the only objective function (see row-$Z_3$ column-$Z_3$).

This value increases to 4,004,945 when minimizing the maximum machine load is the only objective (see row-$Z_8$ column $Z_3$). The last two rows of Table 10 show the best and the worst observed values of each objective function term. From these rows, we can see a considerably large gap between the best and the worst values of an objective function term. The best value of an objective function term is obtained when it is the only term optimized, and the worst is found when it is unaccounted for optimization. These significantly large deteriorations of unaccounted objective function terms in a large size problem greatly emphasize the need for multi-objective optimization in real industrial scheduling problems that are usually large in size.

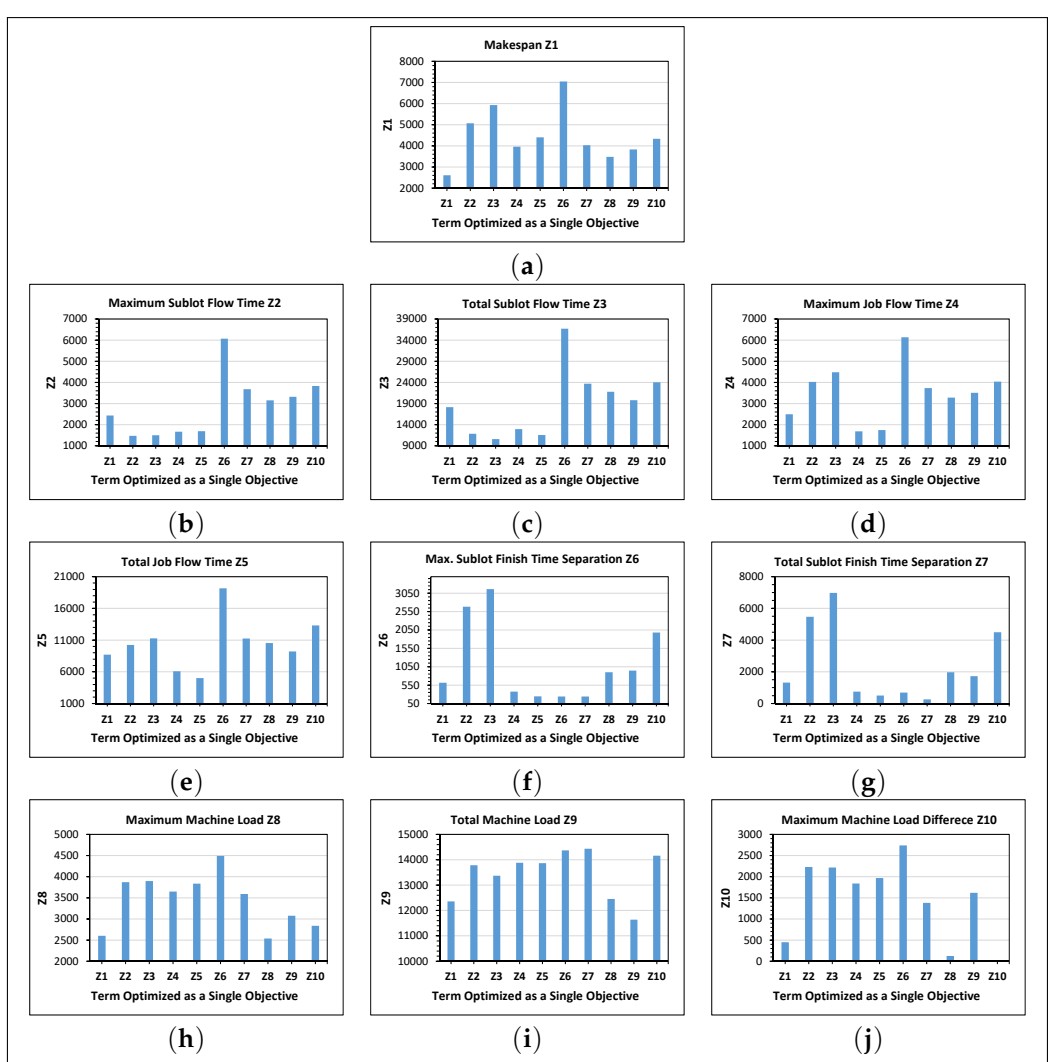

**Figure 9.** Values of the objective function terms in Problem-1 when only one objective function term is optimized.

**Table 10.** Values of the objective function terms in Problem-4 when only one objective function term is optimized.

| Term Optimized as a Single | | Objective Function Term Evaluated | | | | | | | | | |
|---|---|---|---|---|---|---|---|---|---|---|---|
| Objective Function | | $Z_1$ | $Z_2$ | $Z_3$ | $Z_4$ | $Z_5$ | $Z_6$ | $Z_7$ | $Z_8$ | $Z_9$ | $Z_{10}$ |
| Makespan | $Z_1$ | 26,604 | 26,511 | 2,500,084 | 26,594 | 1,002,941 | 8580 | 96,070 | 26,593 | 660,550 | 715 |
| Maximum Sublot Flowtime | $Z_2$ | 28,756 | 24,014 | 2,495,091 | 28,367 | 1,016,441 | 9426 | 110,413 | 28,175 | 665,522 | 3063 |
| Total Sublot Flowtime | $Z_3$ | 28,822 | 27,596 | 2,115,095 | 27,977 | 951,313 | 13,437 | 173,636 | 27,964 | 667,816 | 2747 |
| Maximum Job Flowtime | $Z_4$ | 28,619 | 25,156 | 2,520,010 | 25,161 | 990,089 | 8029 | 71,921 | 28,155 | 663,042 | 2989 |
| Total Job Flowtime | $Z_5$ | 27,316 | 26,770 | 2,312,532 | 26,846 | 868,391 | 6431 | 41,727 | 27,071 | 652,393 | 2169 |
| Maximum Sublot Finish-Time Separation | $Z_6$ | 37,849 | 37,323 | 3,417,121 | 37,608 | 1,337,687 | 709 | 19,222 | 30,936 | 695,833 | 6275 |
| Total Sublot Finish-Time Separation | $Z_7$ | 30,966 | 30,475 | 2,857,497 | 30,561 | 1,110,654 | 1788 | 10,813 | 29,629 | 690,055 | 4700 |
| Maximum Machine Load | $Z_8$ | 44,197 | 43,469 | 4,004,945 | 43,658 | 1,586,503 | 14,786 | 158,042 | 25,793 | 643,747 | 227 |
| Total Machine Load | $Z_9$ | 43,242 | 41,933 | 3,926,277 | 42,356 | 1,494,746 | 15611 | 76,124 | 28,123 | 573,164 | 10,040 |
| Maximum Machine Load Difference | $Z_{10}$ | 36,943 | 36,238 | 3,314,121 | 36,671 | 1,348,060 | 15,639 | 161,671 | 27,729 | 693,034 | 15 |
| Minimum (Best Value) | | 26,604 | 24,014 | 2,115,095 | 25,161 | 868,391 | 709 | 10,813 | 25,793 | 573,164 | 15 |
| Maximum (Worst Value) | | 44,197 | 43,469 | 4,004,945 | 43,658 | 1,586,503 | 15,639 | 173,636 | 30,936 | 695,833 | 10,040 |

### 5.1.3. Jointly Optimizing $Z_1, \ldots, Z_{10}$

In the previous section, the best and the worst values of the various objective function terms were determined when only one term was optimized at a time. In this section, we attempted to illustrate the ability of the proposed algorithm to jointly optimize all the terms and achieve values close to their best-known ones. In doing so, we first provide a plot of the values of the various objective function terms of Problem-4 in Figure 10a when makespan

is the only objective function optimized. In this figure, the values of the objective function terms are plotted on a scale between 0 and 1, corresponding to the best and worst values, respectively.

Makespan achieves its minimum value since it is the only objective optimized. However, from this plot, one can see that several objective function terms, namely $Z_6$, $Z_7$, and $Z_9$, are not close enough to their respective best values obtained when each one of them was the only objective optimized. Next, we solve the same problem to jointly optimize all the objective function terms with equal weights set at one. The resulting values of the objective function terms are plotted in Figure 10b. In this plot, except for $Z_9$, all the values of the objective function terms are close to their best values and far from their worst values.

Finally, Problem-4 was optimized with increased weight for $Z_9$, and the values of terms are plotted in Figure 10c. From this final plot, one can see that all the terms of the objective function are much closer to their best values than to their worst values. This result demonstrates the ability of the proposed algorithm to jointly optimize all the objective function terms considered in the proposed model.

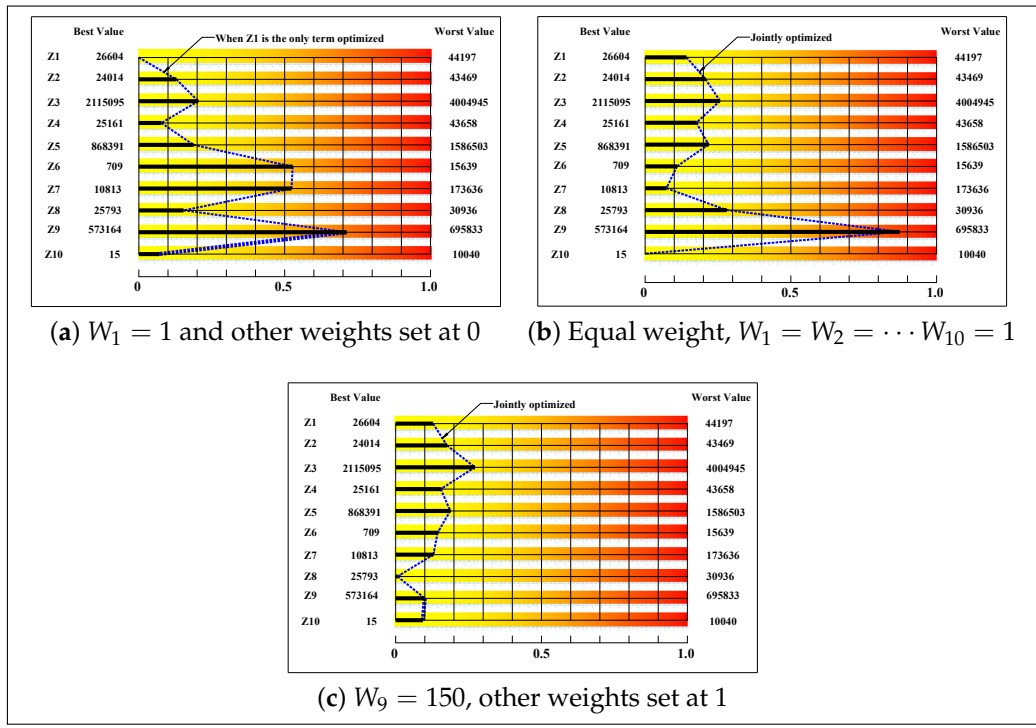

(**a**) $W_1 = 1$ and other weights set at 0     (**b**) Equal weight, $W_1 = W_2 = \cdots W_{10} = 1$

(**c**) $W_9 = 150$, other weights set at 1

**Figure 10.** Values of the objective function terms when (**a**) only makespan $Z_1$ is optimized, (**b**) all terms are jointedly optimized with equal weights set at one, and (**c**) all terms optimized with $W_9 = 150$ and other weights set at one.

### 5.1.4. Further Empirical Study of Objective Functions

In this section, we conducted additional empirical investigations to illustrate the interaction of the objective function terms and their relevance in providing good quality solutions. A total of eleven cases were investigated. The cases differ by the values of the weights of the objective function terms. The settings for the weights for these eleven cases are given in Table 11. In each case, the genetic algorithm was executed ten times, and the average values of the objective function terms were collected. Table 12 provides these values.

**Table 11.** Values of the weights of the objective function terms in Problem-4 in eleven different cases.

| Case | Objective Function Term Weight | | | | | | | | | |
|---|---|---|---|---|---|---|---|---|---|---|
| | $W_1$ | $W_2$ | $W_3$ | $W_4$ | $W_5$ | $W_6$ | $W_7$ | $W_8$ | $W_9$ | $W_{10}$ |
| 0 | 1 | 1 | 1 | 1 | 1 | 1 | 1 | 1 | 1 | 1 |
| 1 | 1 | 1 | 1 | 0 | 0 | 0 | 0 | 1 | 1 | 1 |
| 2 | 1 | 0 | 0 | 1 | 1 | 0 | 0 | 1 | 1 | 1 |
| 3 | 1 | 1 | 1 | 0 | 0 | 1 | 1 | 1 | 1 | 1 |
| 4 | 1 | 0 | 0 | 1 | 1 | 1 | 1 | 1 | 1 | 1 |
| 5 | 1 | 1 | 0 | 0 | 0 | 0 | 0 | 1 | 1 | 1 |
| 6 | 1 | 0 | 1 | 0 | 0 | 0 | 0 | 1 | 1 | 1 |
| 7 | 1 | 0 | 0 | 1 | 0 | 0 | 0 | 1 | 1 | 1 |
| 8 | 1 | 0 | 0 | 0 | 1 | 0 | 0 | 1 | 1 | 1 |
| 9 | 1 | 1 | 1 | 1 | 1 | 1 | 1 | 1 | 0 | 0 |
| 10 | 1 | 1 | 1 | 1 | 1 | 1 | 1 | 0 | 1 | 0 |

**Table 12.** Average values of the objective function terms in Problem-4 from ten replications in each cases (Cases 0 to 10).

| Case | Objective Function Terms | | | | | | | | | |
|---|---|---|---|---|---|---|---|---|---|---|
| | $Z_1$ | $Z_2$ | $Z_3$ | $Z_4$ | $Z_5$ | $Z_6$ | $Z_7$ | $Z_8$ | $Z_9$ | $Z_{10}$ |
| 0 | 24,550 | 23,710 | 1,057,020 | 23,762 | 822,152 | 425 | 2030 | 22,909 | 565,394 | 559 |
| 1 | 24,524 | 23,684 | 1,017,100 | 24,206 | 829,235 | 4642 | 15,476 | 22,657 | 561,306 | 366 |
| 2 | 24,640 | 23,964 | 1,143,060 | 24,021 | 823,195 | 3222 | 14,307 | 23,047 | 571,702 | 372 |
| 3 | 24,891 | 24,183 | 1,053,079 | 24,286 | 847,180 | 375 | 1337 | 22,717 | 562,619 | 396 |
| 4 | 24,789 | 24,322 | 1,127,425 | 24,355 | 835,824 | 579 | 2680 | 22,959 | 569,595 | 376 |
| 5 | 24,503 | 23,470 | 1,195,014 | 24,297 | 877,040 | 5370 | 23,064 | 23,073 | 573,483 | 285 |
| 6 | 24,737 | 24,578 | 1,013,725 | 24,618 | 831,675 | 4550 | 14,209 | 22,639 | 561,504 | 308 |
| 7 | 24,406 | 23,822 | 1,176,456 | 23,839 | 874,850 | 4198 | 17,824 | 22,977 | 571,159 | 258 |
| 8 | 24,546 | 24,447 | 1,088,981 | 24,476 | 815,426 | 4218 | 15,152 | 22,872 | 568,343 | 289 |
| 9 | 24,842 | 23,935 | 998,845 | 23,950 | 811,569 | 84 | 376 | 22,987 | 562,417 | 2047 |
| 10 | 24,756 | 23,870 | 958,741 | 23,881 | 804,054 | 94 | 319 | 23,844 | 552,727 | 4202 |

The setup load is the portion of the total workload $Z_9$ required to perform setup operations.

Case-1 and Case-2 were considered to investigate flowtime performance measures. Case-1 attempts to minimize the maximum and total sublot flowtime ($Z_2$ and $Z_3$), whereas Case-2 attempts to minimize the maximum and total job flowtime ($Z_4$ and $Z_5$). The objective function terms $Z_1$, $Z_8$, $Z_9$ and $Z_{10}$ are also optimized. In shifting from Case-1 to Case-2, the total job flowtime ($Z_5$) changes from 829,235 to 823,195 (less than 1% improvement).

However, the total sublot flowtime ($Z_3$) changes from 1,017,100 to 1,143,060 (12% deterioration). Moreover, Case-2 increased the total workload ($Z_9$) by 10,396 min (a change from 561,306 to 571,702 min). Hence, optimizing the sublot flowtime is more desirable than optimizing job flowtime. However, as it can be seen from the values of $Z_2$, $Z_3$, $Z_4$, and $Z_5$ in Case-0, optimizing both the sublot and job flowtime simultaneously can result in a favorable solution with respect to the overall flowtime performance.

In both Case-1 and Case-2, the maximum and total sublot finish-times separations ($Z_6$ and $Z_7$, respectively) are significant compared to Case-3 and Case-4. Case-3 and Case-4 are similar to Case-1 and Case-2, respectively. However, in these two cases, $Z_6$ and $Z_7$ were also minimized. As can be seen from the result, $Z_6$ and $Z_7$ were reduced substantially with minimal impacts on sublot and job flowtime perforce measures. The result confirms the importance of minimizing sublot finish-time separation along with sublot and job flowtime, which is initially reported in this paper.

Another observation from the empirical study in this section is the importance of jointly minimizing the maximum and the total of a performance measure. For instance, let us examine Case-1, Case-5, and Case-6. In Case-1, both the maximum ($Z_2$) and total ($Z_3$) sublot flowtimes are minimized. Case-5 minimizes $Z_2$ but not $Z_3$, and Case-6 minimizes $Z_3$ but not $Z_2$. The values of ($Z_2$ and $Z_3$) in Case-1, Case-5, and Case-6 are (23,684 and

1,017,100), (23,470 and 1,195,014), and (24,578 and 1,013,725), respectively. In shifting from Case-5 to Case-6, $Z_2$ deteriorates by 4.7%, and $Z_3$ improves by 15%.

Thus, minimizing $Z_2$ alone results in an unfavorable value of $Z_3$ and vice versa. By adopting Case-1, $Z_2$ deteriorates only by 0.9% from its value in Case-5, and $Z_3$ deteriorates only by 0.33% from its value in Case-6. Thus, instead of minimizing the maximum or the total sublot flowtime alone, it is preferable to minimize both of them simultaneously. By examining Case-2, Case-7, and Case-8, we can also arrive at a similar conclusion regarding $Z_4$ and $Z_5$.

In the literature, workload balancing in FJSP has been handled either by minimizing the workload of the most loaded machine (maximum workload $Z_8$) or by minimizing the total workload ($Z_9$). Accordingly, Case-9 minimizes $Z_8$, and Case-10 minimizes $Z_9$. However, in both cases, we can see that the difference between the workloads of the most loaded and the least loaded machines (maximum workload difference, $Z_{10}$) is significant compared to all the cases from Case-1 to Case-8 where $Z_{10}$ is also minimized along with other objective function terms. Thus, for better workload balancing, it is desirable to minimize $Z_{10}$ along with $Z_8$ and $Z_9$. The minimization of $Z_{10}$ to improve workload balancing is reported for the first time in this paper.

### 5.2. Performance Evaluation of RGA and 2SGA

5.2.1. Initial Solution Quality

Bajer et al. [44] and Rahnamayan et al. [45] argued that the quality of the initial population is an important factor in determining the abilities of evolutionary algorithms to find acceptable solutions with minimal execution times. With this in mind, Defersha and Rooyani [1] illustrated that one of the key factors for the success of their two-stage GA is its ability to find initial solutions with greatly improved makespan. In this paper, we further illustrate the ability of 2SGA in finding an improved initial population not only with respect to the makespan but also with many other performance metrics of the multi-objective FJSP lot streaming presented in Section 3.2.

Table 13 provides the means and the standard deviations of the objective functions $Z_1$ to $Z_{10}$ in the initial population of 2000 individuals in Problem-1 and Problem-4. From this table, it can be clearly seen that the mean and the standard deviation of values of the various objective functions in the initial population are greatly improved as we move from RGA to 2SGA. For instance, the mean and standard deviation of the maximum sublot flowtime ($Z_2$) improve by 41% and 57%, respectively, in Problem-1 and by 47% and 79%, respectively, in Problem-4. The histogram for the weighted sum of all the objective function terms of the initial population is displayed in Figure 11. The histogram shows that 2SGA results in highly improved initial solution quality in solving the proposed multi-objective FJSP lot streaming problem.

**Table 13.** The mean and standard deviation of the objective function terms in the initial population under RGA and 2SGA.

| Objective Term | Problem-1 | | | | | Problem-4 | | | | |
|---|---|---|---|---|---|---|---|---|---|---|
| | Mean | | StDev | | Percentage | Mean | | StDev | | Percentage |
| | RGA | 2SGA | RGA | 2SGA | Improvement * | RGA | 2SGA | RGA | 2SGA | Improvement * |
| $Z_1$ | 6317 | 3767 | 972 | 412 | (40, 58) | 58,513 | 31,277 | 3293 | 748 | (47, 77) |
| $Z_2$ | 5477 | 3221 | 1045 | 447 | (41, 57) | 57,435 | 30,524 | 3336 | 690 | (47, 79) |
| $Z_3$ | 30,138 | 18,243 | 6116 | 2422 | (39, 60) | 4,574,999 | 2,476,978 | 299,643 | 89,360 | (46, 70) |
| $Z_4$ | 5895 | 3483 | 1003 | 438 | (41, 56) | 57,957 | 30,846 | 3312 | 705 | (47, 79) |
| $Z_5$ | 18,954 | 10,829 | 3420 | 1035 | (43, 70) | 2,081,854 | 1,122,475 | 119,664 | 17,273 | (46, 86) |
| $Z_6$ | 2896 | 1491 | 1171 | 581 | (49, 50) | 21,250 | 10,607 | 4788 | 2282 | (50, 52) |
| $Z_7$ | 6297 | 3084 | 2772 | 1232 | (51, 56) | 248,166 | 116,497 | 40,929 | 15,422 | (53, 62) |
| $Z_8$ | 5058 | 3336 | 894 | 291 | (34, 67) | 37,728 | 29,216 | 2621 | 538 | (23, 79) |
| $Z_9$ | 14,816 | 13,835 | 596 | 550 | (7, 8) | 682,856 | 667,035 | 6709 | 6442 | (2, 4) |
| $Z_{10}$ | 3789 | 1197 | 1228 | 532 | (68, 57) | 19,471 | 5945 | 3411 | 1183 | (69, 65) |

\* The percentage improvement in Mean and StDev (Mean, StDev) in the initial population achieved by 2SGA.

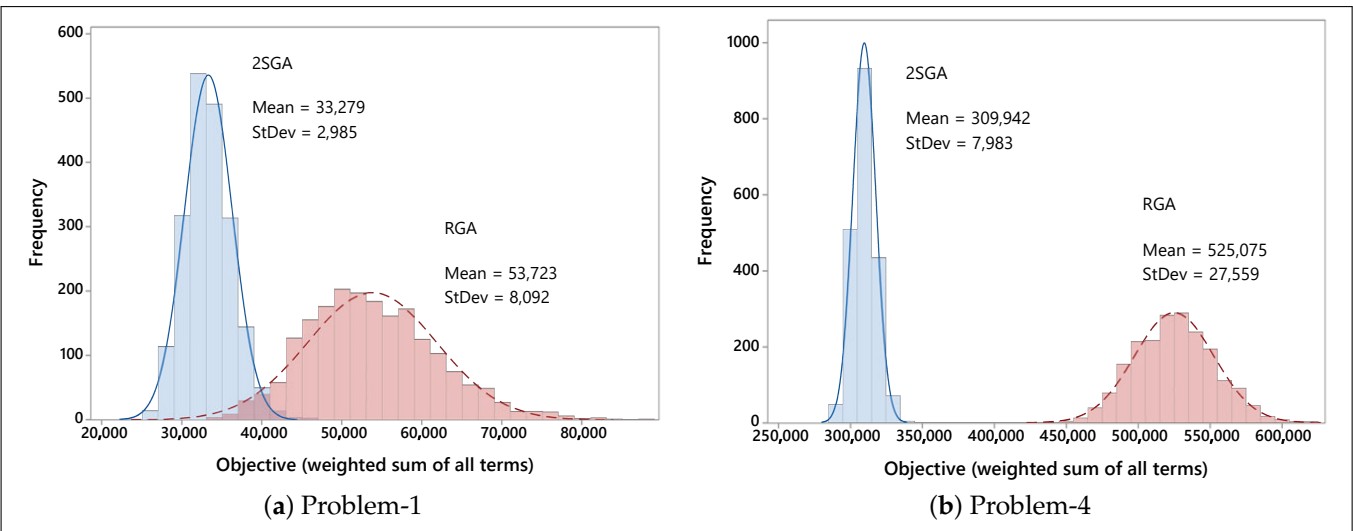

**Figure 11.** The distribution of the objective function of the initial populations of 2SGA and RGA in both Problem-1 and Problem-4.

5.2.2. Convergence Behaviors

The previous section illustrated that 2SGA resulted in an improved initial population in all the objective function terms. In this section, we compare the convergence behavior of 2SGA and RGA while solving large-size problems (Problems 4 to 7). The basic features of these problems are given in Table 14. The parameters of the GAs used in this numerical example are given in Table 15. Figure 12a–g shows the convergence along the objective function terms $Z_1$ to $Z_{10}$, respectively, of 2SGA and RGA in solving Problem-4 while all these terms are simultaneously optimized with equal weight ($W_1 = W_2 = \cdots = W_{10} = 1$). Each convergence curve is an average of 40 replications.

**Table 14.** Basic features of the problems considered for performance evaluation of the proposed algorithm.

| Problem | $M$ | $J$ | $S_j$ (max) | $O_j$ (min, max) | NAMPJ (min, max) * |
|---|---|---|---|---|---|
| 4 | 25 | 40 | 4 | (8 15) | (3, 6) |
| 5 | 30 | 60 | 4 | (8, 16) | (3, 6) |
| 6 | 40 | 80 | 4 | (10, 18) | (2, 8) |
| 7 | 50 | 100 | 4 | (10, 20) | (2, 8) |

* NAMPJ = Number of Alternative Routing per Operation.

**Table 15.** Algorithm parameters.

| Parameters | Values |
|---|---|
| Population Size | 2000 |
| Tournament Size Factor $\alpha$ | 0.005 |
| Crossover Probability | 0.85 |
| Mutation Probability | 0.15 |
| Number of generation for the first sage in 2SGA | 2500 |
| Total number of genration | 10,000 |
| $W_1, W_2, \cdots, W_{10}$ | 1.0 |

Note: Tournament size = $\alpha \times$ Population size.

From these convergence curves, we can see that 2SGA was able to converge more rapidly than RGA along $Z_1$ to $Z_5$, $Z_8$, and $Z_9$. In terms of these objective function terms, 2SGA was able to find better solutions in only a few hundred generations compared with those determined after more than 10,000 generations by RGA. In terms of $Z_6$, $Z_7$, and $Z_{10}$, RGA was able to converge more rapidly than 2SGA. However, 2SGA was able to catch up with RGA only in a few hundreds of generations right after it changed the search stage, which occurred at 2500 generations. Figure 12h is the convergence of 2SGA and RGA in

terms of the weighted sum of all the objective function terms, which clearly shows the superiority of 2SGA over RGA. From the convergence graphs, the first stage of 2SGA was able to achieve convergence within the first few hundreds of generations.

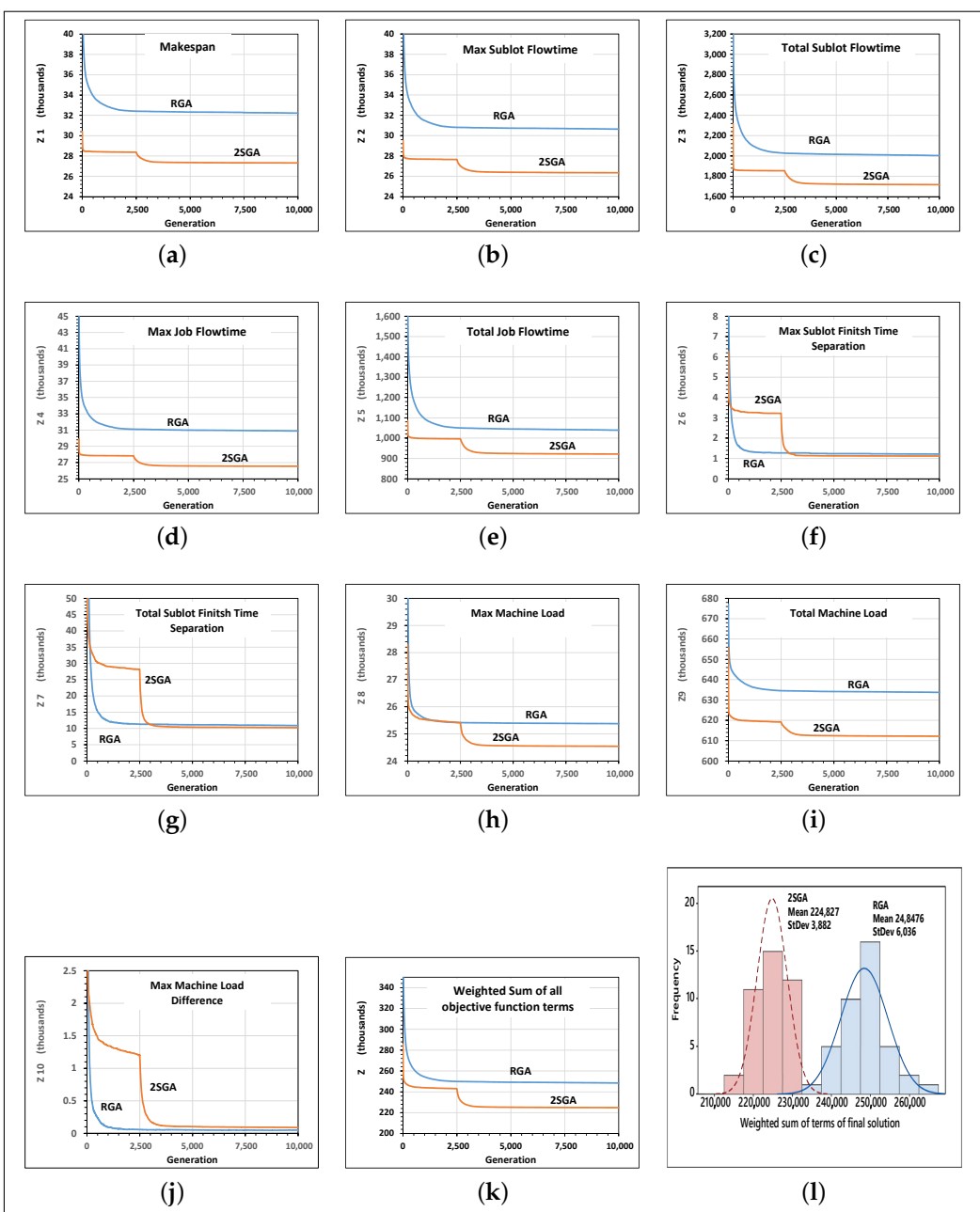

**Figure 12.** The convergence of 2SGA and RGA in solving Problem 4 while all the objective terms are optimized simultaneously. (Each convergence graph (**a**–**k**) is an average of 40 trials, and (**l**) is the histogram of the final values of objective function in these 40 trials).

For instance, if 2SGA changed its search stage at 1000 generation, it could provide highly improved solutions in only 3000 generations, which cannot be achieved using RGA after many thousands of generations. Figure 12i depicts the histograms of the objective function of the final solutions in 40 trials in both 2SGA and RGA. In these histograms, 2SGA achieves approximately 9.5% and 35.6% improvements in the mean and standard deviation, respectively. An improvement in the standard deviation by 2SGA represents its robustness in finding good solutions more consistently than RGA. Similar results were obtained while solving Problems-5, 6, and 7, as shown in Figure 13. The computational times required by

2SGA and RGA to complete the 10,000 generations using the parameters in Table 15 were approximately 120, 335, 840, and 1410 min in Problems 4, 5, 6, and 7, respectively.

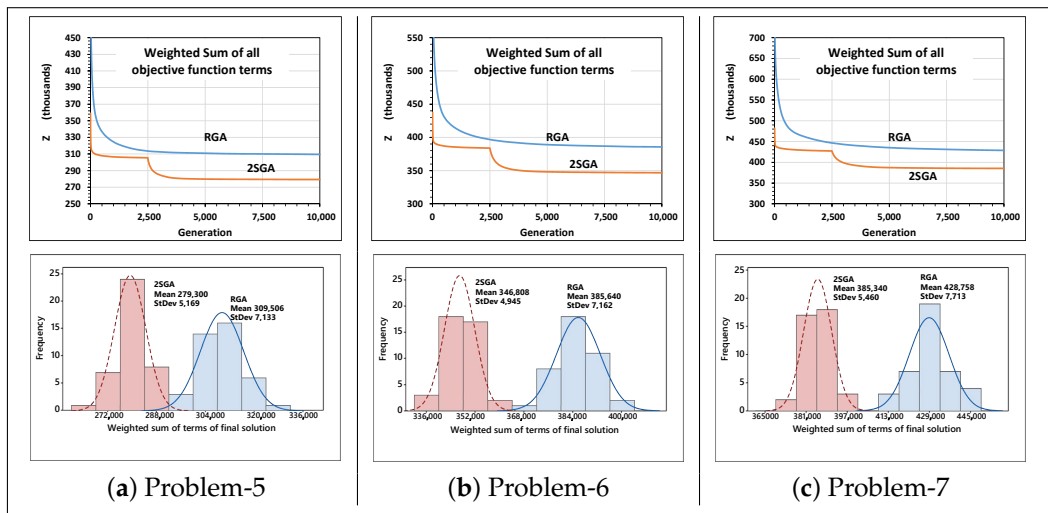

**Figure 13.** The convergence of 2SGA and RGA in solving Problems 4, 5, and 6. (Each convergence graph is an average of 40 trials. The histograms are for the final values of the objective function in 40 trials).

### 5.2.3. Improvement through Parallelization

Parallelizing genetic algorithms using a high-performance parallel computing platform has been well recognized as a viable technique to enhance their abilities in solving many complex and large-size problems. Its application in solving shop scheduling problems has also been widely reported as reviewed in [46]. In this paper, we adopted a randomly connected multi-population parallel GA (P-GA) proposed in [47] to illustrate the performance improvement that can be achieved in both RGA and 2SGA. The P-GA consists of several subpopulations where each of them is assigned to a dedicated CPU.

A subpopulation evolves independently and communicates periodically by sending and receiving selected solutions to and from other subpopulations. Whenever communication occurs, the CPU with rank 0 randomly generates a communication matrix and broadcasts it to all other CPUs. The migration of the copies of the selected solutions follows the route generated according to the communication matrix. An example communication matrix and the resulting migration route for a small instance of parallelization are depicted in Figure 14 where the CPUs are ranked from 0 to 6. The density of the communication matrix, the frequency of communication, and the strategy for the selection and replacement of migrants from the source and to the destination subpopulations are key parameters for this parallelization technique. An investigation of these parameters is not within the scope of this paper.

In this study, we used a total of 80 concurrently available CPUs in high performance parallel computing platform to implement the parallel RGA (P-RGA) and the parallel 2SGA (P-2SGA). Problems 4, 5, 6, and 7 were solved using both the sequential and the parallel versions of these algorithms using a subpopulation size of 2000. The subpopulations were allowed to communicate every 30 generations. The change of stage for 2SGA occurred at 2500 generations. The computation was terminated after 10,000 generations. The resulting convergence graphs are given in Figure 15. From these graphs, one can see that parallelization brings performance improvements in both RGA and 2SGA. However, the crucial finding in this investigation is that the sequential 2SGA using a single CPU outperforms the parallel RGA that uses 80 CPUs. This finding asserts the superiority of 2SGA over RGA in solving the proposed multi-objective FJSP lot streaming problem.

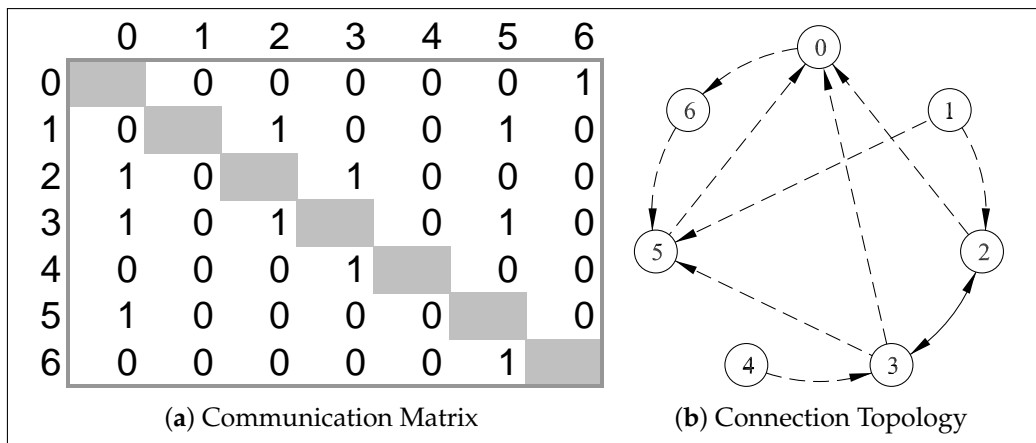

**Figure 14.** Randomly connected topologies for a given communication matrices (adopted from [47]). Note: The communication matrix and topology is generated every time before solution migration occurs.

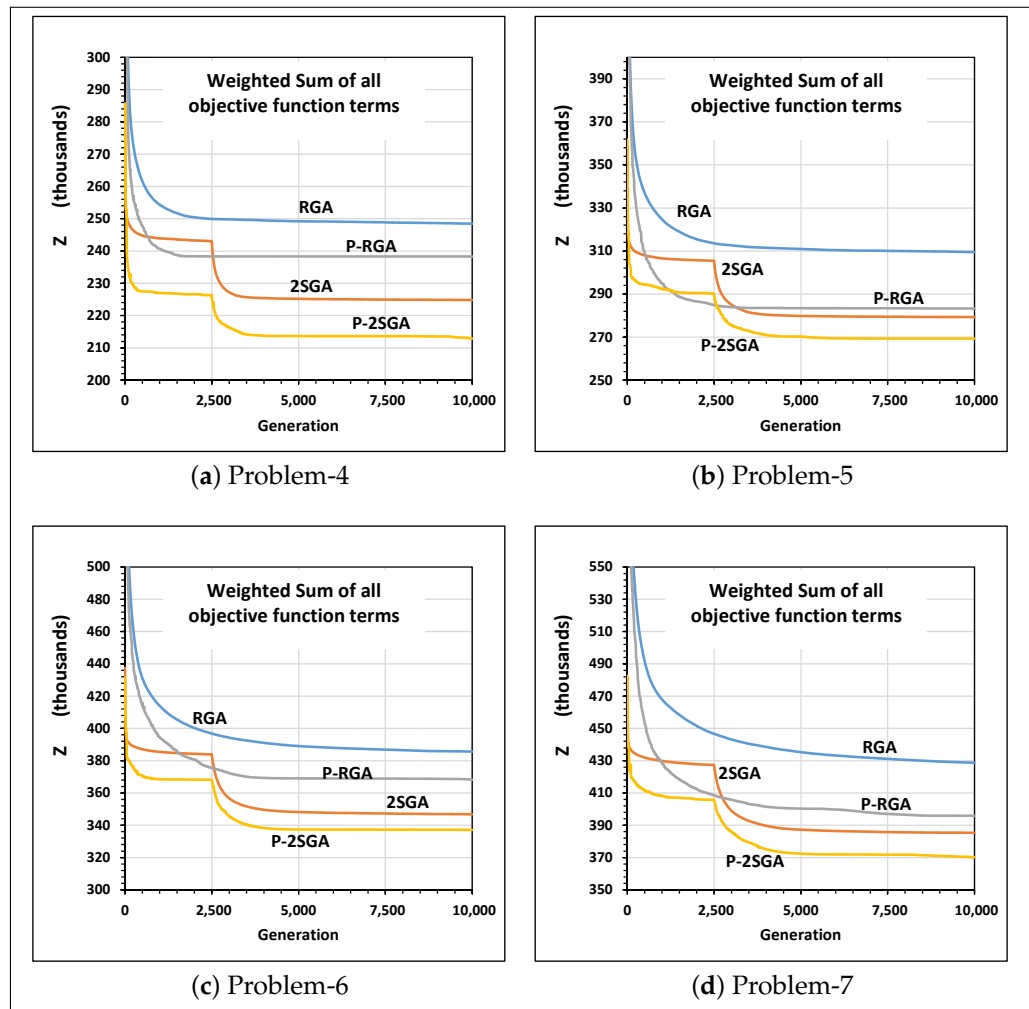

**Figure 15.** Improvements of convergence behaviors of RGA and 2SGA using the parallelization technique.

### 5.3. Empirical Analysis of the Algorithm Parameters

5.3.1. Selection Operators

In this section, we present comparative empirical studies on the various selection operators presented in Section 4.5. The comparisons are presented in terms of the convergence behavior of 2SGA in solving Problem 4, whereas similar results were obtained in solving several other problems. The first of these imperial studies is aimed at comparing the three fitness transformation functions in Equations (60)–(62) used in the proportional selection method. Figure 16 provides the average convergence from ten test runs using these three different fitness transformation equations. As can be seen from this figure, Equations (60) and (61) resulted in similar convergence behaviors of the algorithm. In contrast, the transformation function in Equation (62) resulted in a much better convergence of 2SGA when using the proportional selection method.

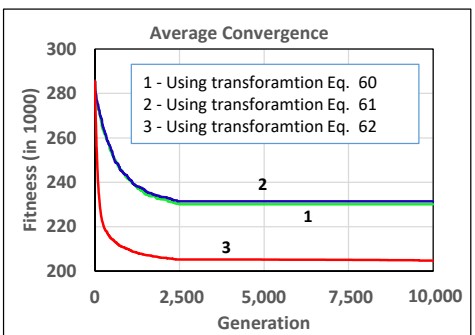

**Figure 16.** The average convergence from ten test runs of 2SGA using proportional selection under three different fitness transformation while solving Problem-4.

The second empirical study investigates the impact of tournament size in tournament selection. Figure 17 depicts the results of this study. This figure shows that tournament selection with a smaller tournament size was preferred in solving the proposed mathematical model using 2SGA. Lastly, a comparison of proportional, linear ranking, and tournament selection was conducted, and the resulting convergence graphs are given in Figure 18. This figure shows that the tournament selection resulted in an improved convergence of the proposed 2SGA. Hence, tournament selection with a small tournament size is the preferred selection operator in the proposed algorithm.

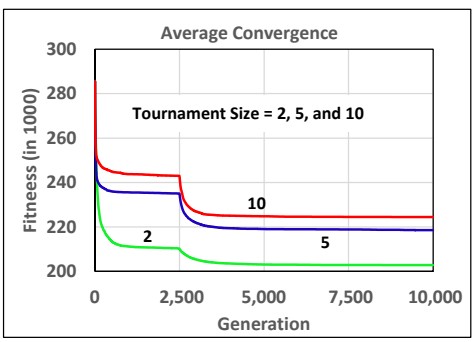

**Figure 17.** The average convergence from ten test runs of 2SGA using tournament selection under different tournament sizes while solving Problem-4.

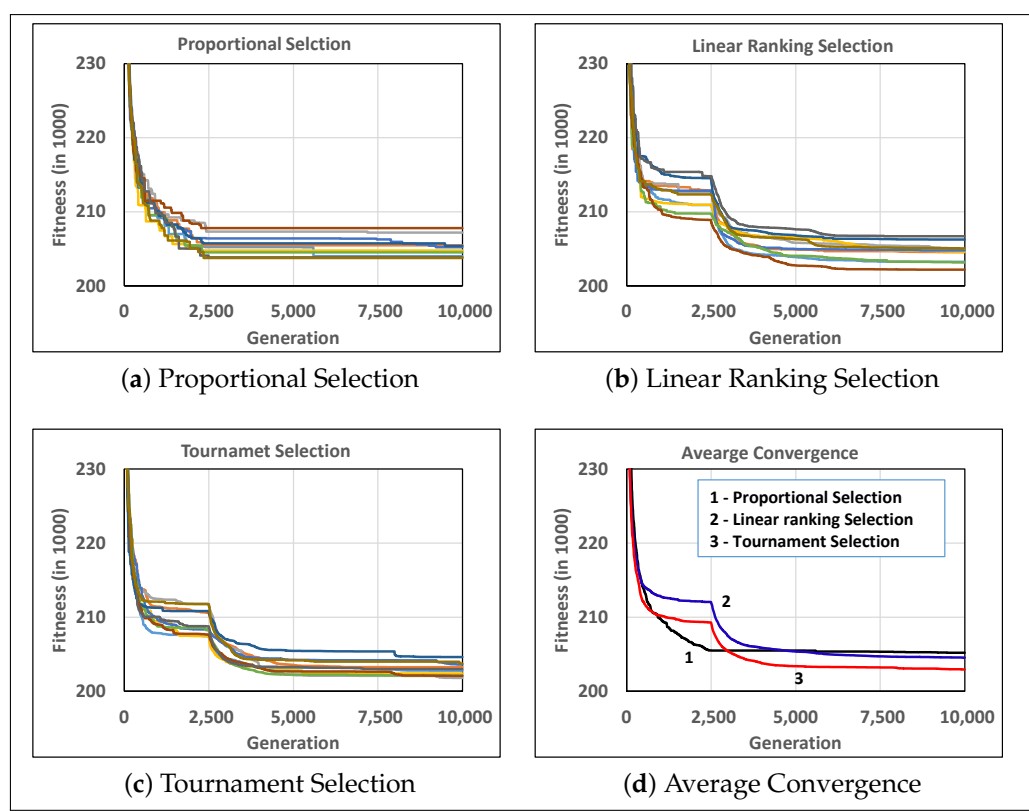

**Figure 18.** The convergence of 2SGA under three different selection operators while solving Problem-4.

### 5.3.2. Crossover and Mutation Probabilities

In the proposed algorithm, there are seven crossover and six mutation operators. Assigning probabilities individually for these thirteen operators and simultaneously tuning them can be a daunting task. Instead, in this paper, we suggested the crossover and mutation operators be assigned one crossover and one mutation probability, respectively. With this scheme, we performed an Analysis of Variance (ANOVA), where mutation and crossover probabilities are the only two factors, and the objective function is the response. We chose six levels for each of these factors. The levels for the mutation and crossover probabilities were {0.05, 0.15, 0.25, 0.35, 0.45, 0.55} and {0.75, 0.80, 0.85, 0.90, 0.95, 1.00}, respectively. For each factor level combination, we conducted five replications. Hence, the experiment required solving a problem 180 times. The genetic algorithm used a different seed for its random number generator in each replication of the experiment.

The results of ANOVA for Problem-4 are presented in Table 16 and Figure 19. The *p*-values corresponding to the main effects of mutation and crossover probabilities are zero, thus, implying that these two factors have statistically significant effects on the final solution quality. On the other hand, the *p*-value for the interaction effect is high (compared to a typical significance level $\alpha = 0.05$), which indicates the absence of interaction between these two factors. This lack of interaction simplifies parameter tuning, thereby, allowing the user to optimize them independently.

The plots of the main effects in Figure 19 show that the mutation probability needs to be set close to 0.35, and the crossover needs to be set at higher values between 0.90 and 1.00. The residual plots do not indicate unusual patterns, thus, confirming the adequacy of the ANOVA. The analysis also rendered similar results on several other problems of a varying size considered in this paper. Hence, the recommended values of the mutation and crossover probabilities can be used to solve different sets of problems using the proposed algorithm.

**Table 16.** Output of the analysis of variance for Problem-4.

| Source | DF | Adj SS | Adj MS | F-Value | *p*-Value |
|---|---|---|---|---|---|
| Mutation Probability | 5 | 1,418,299,280 | 283,659,856 | 93.7 | 0.000 |
| Crossover Probability | 5 | 91,944,634 | 18,388,927 | 6.03 | 0.000 |
| Mutation Probability*Crossover Probability | 25 | 49,082,335 | 1,963,293 | 0.64 | 0.901 |
| Error | 144 | 438,889,368 | 3,047,843 | | |
| Total | 179 | 1,998,215,617 | | | |

DF = Degrees of Freedom; Adj SS = Adjusted sum of square; and Adj MS = Adjusted mean square.

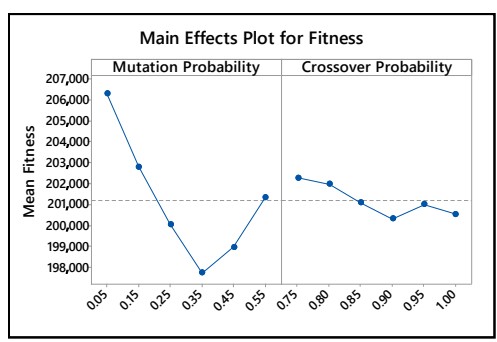
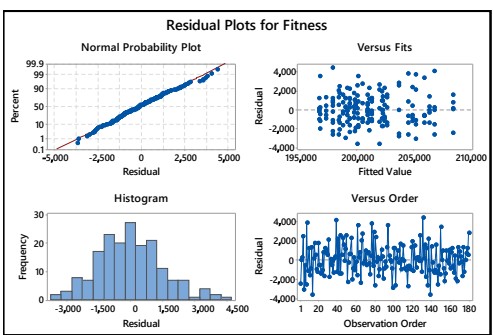

**Figure 19.** The main effect and residual plots of the analysis of variance.

## 6. Discussion, Conclusions, and Future Research

### 6.1. Discussion and Conclusions

The recent trend in manufacturing scheduling is in developing efficient algorithms for complex and comprehensive scheduling problems. Apparently following this trend, a group of authors recently developed an efficient two-stage genetic algorithm for a comprehensive flexible job shop scheduling problem (FJSP) that incorporated (1) sequence-dependent setup time, (2) attached and detached nature of setups, (3) machine release date, and (4) lag-time. The authors demonstrated the superiority of the developed two-stage genetic algorithm in solving large-size problems, which motivated our current research. In this paper, we expand the application of the two-stage genetic algorithm to solve a comprehensive flexible job shop lot streaming problem that incorporates many objective functions. Several empirical investigations were conducted on the proposed model and the two-stage genetic algorithm through which the following observations and conclusions were made.

- The magnitude of the severity of a single objective optimization on the objective function terms that are not incorporated increased as the problem size increased. The result emphasizes the need for multi-objective optimization in real industrial scheduling problems that are typically large in size.
- Optimizing the sublot flowtime is more desirable than optimizing the job flowtime. However, optimizing both terms simultaneously can also result in favorable solutions with respect to the overall flowtime performance.
- In lot streaming, one sublot of a given job may be finished much sooner than the other sublot of the same job. This may increase the work-in-process inventory. The newly proposed objective function terms (to minimize the maximum sublot finish-time separation and total sublot finish-time separation) can alleviate this problem with minimal impacts on the sublot and job flowtime.
- Instead of minimizing the maximum or the total sublot flowtime, it is advantageous to minimize both its maximum and total values simultaneously. The same is true with the other performance measures (the job flowtime, sublot finish-time separation, and machine workload).
- Workload balancing in FJSP may not be fully achieved by minimizing the maximum or the total workload or both. A newly proposed objective function term (minimizing

the maximum workload difference), can result in a better workload balance when considered along with the minimization of the maximum and/or the total workload.

- The solution representation and the corresponding decoding of the first stage of the two-stage genetic algorithm can generate initial solutions that are highly improved in all the ten objective function terms.
- The two-stage genetic algorithm can jointly optimize all the ten objective function terms of the multi-objective FJSP lot streaming considered in this paper and greatly outperform the regular genetic algorithm.
- Parallel computation can bring performance improvements in both the two-stage GA and the regular GA. However, the crucial finding is that the sequential two-stage GA using a single CPU outperformed the parallel regular genetic algorithm that uses many CPUs in solving the proposed multi-objective FJSP lot streaming problem.
- The performance of the proportional selection method can be significantly improved by the appropriate choice of the fitness transformation function.
- Both proportional, linear ranking and tournament selection can result in comparable performance. However, tournament selection with smaller size of tournament slightly outperformed the other two.
- Analysis of variance shows the lack of interaction between mutation and crossover probabilities. Thus, the two probabilities can be tuned independently.

*6.2. Future Research*

The two-stage genetic algorithm may not be directly applicable in scheduling problems with the objective of minimizing earliness-tardiness. In particular, the greedy nature of the first stage is based on finding a schedule that finishes the jobs as early as possible, which is against minimizing earliness. For instance, finishing jobs too early may represent excess work-in-process in a JIT environment. Hence, our future research includes the development of a modified two-stage genetic algorithm to incorporate the minimization of earliness. We also plan to expand the application of the two-stage genetic algorithm for a multi-resource constrained FJSP as flexible job shops are often constrained by many kinds of resources in addition to machines.

**Author Contributions:** D.R.: Methodology, Investigation, Validation, Formal analysis, Writing— Original Draft, Data Curation; F.D.: Conceptualization, Methodology, Formal analysis, Software, Validation, Visualization, Writing—Review & Editing, Supervision. All authors have read and agreed to the published version of the manuscript.

**Funding:** This research was funded by the Discovery Grant from NSERC, the Natural Science and Engineering Research Counsel of Canada (https://www.nserc-crsng.gc.ca/index_eng.asp) (accessed on 10 June 2022).

**Data Availability Statement:** Not applicable.

**Acknowledgments:** We would like to thank Compute Canada (https://www.computecanada.ca/ (accessed on 10 June 2022)) for providing access to high performance parallel computation facility through which the majority the numerical studies were conducted. We also would like to thank all three anonymous reviewers for their suggestions that helped us to improve the paper.

**Conflicts of Interest:** The authors declare no conflict of interest.

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
