# Peer review of "A Two-Stage Multi-Objective Genetic Algorithm for a Flexible Job Shop Scheduling Problem with Lot Streaming"

_algorithms, doi:10.3390/a15070246_

Round 1

Reviewer 1 Report

The research methodology is appropriate. The paper reviews the related work well, and the authors reference the publications mainly from the last 10 years concerning GA and their application to solving real industrial scheduling problems. To sum up, the authors present results clearly, and the article is a solid journal paper.

In my opinion, the article requires minor improvements - see the article's weaknesses below.

1) The analysis of the combined consideration of multiple optimization criteria in the considered scheduling problem on the quality of the obtained results concerns only the cases with weights 0-1 (see Table 11). There is no information whether the described properties concerning combining criteria to obtain better results are also preserved in the case of other weights of individual criteria.  

2) The manuscript extends the results of other authors' work. In particular, it expands the single objective flexible job shop scheduling problem lot streaming model presented in [2] to a multi-objective one and develops a two-stage genetic algorithm based on the work [1].

This knowledge leads to the concern that the plagiarism level concerning some previous works of the authors and the total plagiarism level can exceed acceptable levels. It is suggested to check the plagiarism levels.    

3) There are also minor typos and fragments in the paper that need clarification.

List of inaccuracies:

1) Page 5, line 193

 … indexed by r or u=1,2,…,Rm;  …

 It is not clear what this means?

 2) Page 15

 Step 4.

Maybe it is worth adding the notation for the completion time, e.g.,

… machines m, i.e., c_o,s,j,m.

Step 7.

Maybe:

… calculated ** at ** Step 4 corresponding to machine …  

 3) Page 21

 Figure 5.

 (b) SSC-2 . < -- dot is no need

 In Job -2, (b) SSC-2, there is alfa with no index!

 4) Page 39, line 936

 … of the proportional section method …

 Should be:   … of the proportional selection method …

Author Response

Please see the attached file for our response to reviewer #1.

Reviewer 2 Report

This paper presents an authors’  former published genetic algorithm [2], on lot streaming scheduling applied to multiple objectives and in a two-steps way based also on an authors’ former work [1]

The work is properly presented, with a clear redaction, with a comprehensive review of former contributions in different metaheuristics and other programming techniques. The main formulation is reviewed briefly from the former work and the different objective formulations are developed in detail. Also, the encoding process is presented in detail. The trials of performance are also presented with all the detail.

The title is adequate to the content. The abstract is well constructed and maintains proportionality with the paper extension, but it could be shortened to about 200 words as usual. The paper length is high but the explanations are detailed, so this balanced exposition is a main value of the paper that justifies in my opinion the extension.

The main algorithm is part of a former work that cannot be followed completely by this reviewer but is a contribution previously validated. The performance of the two steps algorithm in single and parallel computation is quite a proper contribution to the core topic of the Journal. 

Author Response

Thank you for the very positive comment.

We understood that the abstract is a bit long. However, given the very dense nature of the paper, we believe that the length of the abstract is appropriate.

Reviewer 3 Report

Overall paper is written very well.

The introduction need to be improved showing the motivation of your work, highlighting the research problem, and your contributions in this work point by point.

You need to add 4 to 6 new recent papers. One of the suggested is "1) Bin Waheed, M. H., Jamil, F., Qayyum, A., Jamil, H., Cheikhrouhou, O., Ibrahim, M., ... & Hmam, H. (2021). A new efficient architecture for adaptive bit-rate video streaming. Sustainability13(8), 4541.

Mubeen, A., Ibrahim, M., Bibi, N., Baz, M., Hamam, H., & Cheikhrouhou, O. (2021). Alts: An Adaptive Load Balanced Task Scheduling Approach for Cloud Computing. Processes9(9), 1514."

Author Response

We thank you so much for the positive comment.

However, we believe that the instruction is well written and highlights our contribution. 

The suggested additional references are not very relevant.